# Comparison of spatial downscaling methods of general circulation model results to study climate variability during the Last Glacial Maximum

Guillaume Latombe[1,2], Ariane Burke[3], Mathieu Vrac[4], Guillaume Levavasseur[5], Christophe Dumas[4], Masa Kageyama[4], Gilles Ramstein[4]

[1]Centre for invasion Biology, Department of Mathematical Sciences, Stellenbosch University, Matieland 7602, South Africa
[2]School of Biological Sciences, Monash University, Melbourne 3800, Australia
[3]Département d'Anthropologie, Université de Montréal, Montréal, QC, Canada
[4]Laboratoire des Sciences du Climat et de l'Environnement/Institut Pierre-Simon Laplace, Université Paris-Saclay, CE Saclay, l'Orme des Merisiers, Bât.701, Gif-sur-Yvette, France
[5]Institut Pierre Simon Laplace (IPSL), Pôle de modélisation du climat, UPMC, Paris, France

*Correspondence to*: Guillaume Latombe (latombe.guillaume@gmail.com)

**Abstract.** The extent to which climate conditions influenced the spatial distribution of hominin populations in the past is highly debated. General Circulation Models (GCMs) and archaeological data have been used to address this issue. Most GCMs are not currently capable of simulating past surface climate conditions with sufficiently detailed spatial resolution to distinguish areas of potential hominin habitat, however. In this paper we propose a Statistical Downscaling Method (SDM) for increasing the resolution of climate model outputs in a computationally efficient way. Our method uses a generalized additive model (GAM), calibrated over present-day climatology data, to statistically downscale temperature and precipitation time series from the outputs of a GCM simulating the climate of the Last Glacial Maximum (19-23,000 BP) over Western Europe. Once the SDM is calibrated, we first interpolate the coarse-scale GCM outputs to the final resolution and then use the GAM to compute surface air temperature and precipitation levels using these interpolated GCM outputs and fine resolution geographical variables such as topography and distance from an ocean. The GAM acts as a transfer function, capturing non-linear relationships between variables at different spatial scales and correcting for the GCM biases. We tested three different techniques for the first interpolation of GCM output: bilinear, bicubic, and kriging. The resulting SDMs were evaluated by comparing downscaled temperature and precipitation at local sites with paleoclimate reconstructions based on paleoclimate archives (archaeozoological and palynological data) and the impact of the interpolation technique on patterns of variability was explored. The SDM based on kriging interpolation, providing the best accuracy, was then validated on present-day data outside of the calibration period. Our results show that the downscaled temperature and precipitation values are in good agreement with paleoclimate reconstructions at local sites, and that our method for producing fine-grained paleoclimate simulations is therefore suitable for conducting paleo-anthropological research. It is nonetheless important to calibrate the GAM on a range of data encompassing the data to be downscaled. Otherwise, the SDM is likely to over-correct the coarse-grain data. In addition, the bilinear and bicubic interpolation techniques were shown to distort either the temporal variability

or the values of the response variables, while the kriging method offered the best compromise. Since climate variability is an aspect of the environment to which human populations may have responded in the past the choice of interpolation technique is therefore an important consideration.

## 1 Introduction

The extent to which past climate change influenced human population dynamics during the course of prehistory is a subject of lively debate. The Last Glacial period, including Marine Isotope Stages 3 (MIS3) and 2 (MIS2) and the Last Glacial Maximum (LGM), is particularly interesting in the context of this debate (van Andel, 2003). During MIS3 the archaeological record suggests that modern human populations originating in Africa expanded into Eurasia, while Neanderthal populations gradually contracted their range before becoming extinct ~27,000 years Before Present (BP) (Serangeli and Bolus, 2008). Progressively

colder and drier conditions, culminating in the LGM (19,000 - 23,000 years Before Present), are thought to have triggered further range contractions and the demographic decline of modern human populations in Europe. Climate affects the spatial behaviour of human populations directly (when conditions exceed human physiological limits) and indirectly (when it affects the distribution of resources upon which humans depend). The global climate during the Last Glacial period was characterised by a series of rapid oscillations (known as Dansgaard-Oeschger, or D-O events). These events may have acted as forcing

mechanisms, affecting the demographic processes described above (e.g. Müller et al., 2011; Schmidt et al., 2012; Sepulchre et al., 2007; Jimenez-Espejo et al., 2007; Banks et al., 2013; d'Errico and Sánchez Goñi, 2003; Gamble et al., 2004; Shea, 2008). While the timing of climate events relative to large-scale patterns in the archaeological record is suggestive, the mechanisms by which climate forcing acted on human populations are still imperfectly understood. More empirical evidence is needed to validate the hypothesis that climate forcing affected human population dynamics and explore the nature and scale of the effect.

The broad demographic patterns mentioned above are the result of smaller, local-scale patterns produced by mobile groups of hunter-gatherers distributing themselves on the landscape in order to exploit available resources. The availability of these resources fluctuated both predictably (on a seasonal basis) and unpredictably (as a result of climate variability). It is by gaining an understanding of these smaller-scale patterns, ultimately, that we will be able to understand how climate forcing affects the

25 spatial and cultural dynamics of prehistoric human populations. Previous analyses of climate forcing have used a variety of data to reconstruct the paleoclimate, such as ice-core or marine records (e.g., Bradtmöller et al., 2012; Jimenez-Espejo et al., 2007; Schmidt et al., 2012), present-day climate data (e.g., Jennings et al., 2011), and climate model simulations (e.g., Banks et al., 2008; Davies and Gollop, 2003; Sepulchre et al., 2007; Benito et al. 2017; Hughes et al. 2007; Tallavaara et al. 2015). These analyses were conducted at varying spatial resolutions, typically on the order of ~50 km x 50 km (= 2500 km$^2$). Higher-

30 resolution climate simulations are nonetheless necessary for the quantification of climate variability at an inter-annual scale and a spatial scale which approximates the size of the catchments within which hunter-gatherer groups typically forage (~10 km from camp, or 314 km$^2$; Vita-Finzi and Higgs, 1970) making this an ideal spatial scale at which to consider the impact of climate variability on human systems.

Global Climate Models (GCMs) are able to simulate climate conditions at various spatial and temporal scales, whereas climate proxy data are inherently limited by the uneven distribution of sample locations and taphonomic biases (i.e biases in the fossil record, such as pollen preservation, location of archaeological sites, etc.). GCMs use physical equations, e.g. to represent atmospheric fluid dynamics, as well as parameterisations, e.g. for sub-grid scale phenomena, to simulate the Earth's climate. The major disadvantage of GCMs is that they are computationally intensive and are usually only used to model climate behaviour at relatively coarse spatial resolution, typically coarser than 100 km (cf. Flato et al., 2013 for the latest details on the CMIP5 models) especially for long paleoclimatic simulations. Their ability to simulate the small-scale physical processes that drive local surface variables, such as precipitation, is therefore limited (Wood et al., 2004).

Regional Climate Models (RCMs) represent a physically based approach to climate modelling at a finer spatial scale over a specific region of interest (*e.g.*, Liang et al., 2006, Flato et al., 2013). However, RCMs use GCM outputs to set their boundary conditions. They therefore require the explicit modelling of the physical processes at both coarse and fine scales over the whole planet and over region of interest, respectively, and are also computationally demanding. Statistical Downscaling Methods (SDM), on the other hand, are less computationally demanding. SDMs proceed by empirically associating local-scale variables with large-scale atmospheric variables produced by GCMs, and are faster to compute than mechanistic RCMs (Vaittinada Ayar et al., 2015). SDMs fall into four main families: "transfer functions", which directly link large-scale and local-scale variables; "weather typing" methods based on conditioning statistical models on recurrent weather states; "stochastic weather generators" that simulate downscaled values from their (potentially conditional) probability density functions; and "Model Output Statistics" (MOS) methods based on adjusting (i.e., correcting) the statistical distribution of the large-scale GCM simulations in order to generate local-scale variables with the correct statistical properties (e.g., Vaittinada Ayar et al., 2015).

In this study, we explore and refine the capacity of an SDM from the transfer functions family, based on Generalized Additive Modelling (GAM), to compute temperature and precipitation time series at a fine spatial and temporal resolution for the LGM over Western Europe, south of the Fennoscandian ice-sheets. GAM is a non-parametric statistical technique that has proven reliable for capturing non-linear relationships between local- and large-scale variables and correcting the biases specific to a given GCM (e.g., Vrac et al., 2007; Levavasseur et al., 2010). The SDM used here accurately downscales the climatology (i.e., the climate averages over several decades) of temperature and precipitation generated by a GCM for the LGM when calibrated using present-day data (Vrac et al. 2007). Its ability to generate projections of the small-scale temporal patterns necessary to explain the spatial dynamics of prehistoric human populations is untested, however. In the present study, therefore, we use present-day climate data (corresponding to the average of the 1961-1990 period) extracted from the IPSL-CM5A-LR GCM (Dufresne et al., 2013) to calibrate the SDM, applying it to a 50 year-long time series of climate simulations for the LGM (Kageyama et al., 2013a, b). Interpolated values of coarse-grain variables extracted from the GCM, as well as fine-scale geographical data such as elevation and advective continentality, are used as predictors in the GAM. The result is the

production of downscaled monthly values over 50 years for temperature and precipitation, including local temporal variability in temperature and precipitation rates. In addition, we compare the impact of three different interpolation techniques (bilinear interpolation, bicubic interpolation and kriging) on the downscaling results, evaluating the resulting SDM outputs with the aid of climate proxies (palynological and archaeozoological data) and observing the impact of each technique on patterns of spatial

and temporal variability in temperature and precipitation. The SDM using kriging interpolation is demonstrated to be a good compromise between computational complexity and accuracy, validated on an 11-year present-day time series distinct from the calibration period.

## 2 Materials and Methods

### 2.1 Global Climate Model

The GCM used in this study is the ocean-atmosphere coupled model IPSL-CM5A-LR (Dufresne et al., 2013) developed for the CMIP5 (Taylor et al., 2012) and PMIP3 (Braconnot et al., 2012) projects and the 5[th] IPCC report (IPCC, 2013). The IPSL-CM5A-LR model has a spatial resolution of 1.9° in latitude and 3.75° in longitude over Europe, which is the area of interest here (i.e. ~62 500 km$^2$). The model performance and main biases are described in Dufresne et al. (2013) and Hourdin et al. (2013). This model version is known to have a cold (-1.4°C) bias in terms of globally averaged temperature and the bias in

mean annual temperature over Europe is similar to the global value. In this low-resolution version of the model, the mid-latitude westerly winds are generally more equatorial than observed (Hourdin et al., 2013) and this is the case for the Northeast Atlantic and Europe too. The general pattern of extra-tropical precipitation over the North Atlantic and European sectors is satisfactory with respect to the annual mean (Dufresne et al., 2013). The equilibrium temperature response to a doubling $CO_2$ is 3.59°C (Dufresne et al., 2013), which is rather high compared to other CMIP5 models. Nevertheless, the model's response

to LGM forcings somewhat underestimates the cooling over Europe, as reconstructed from pollen, and is satisfactory in terms of precipitation (Kageyama et al., 2013).

We use a *historical* simulation run according to the CMIP5 protocol, and use model output for the period from 1961 to 1990 as our present-day reference climate. Outputs from this simulation are used in the calibration process (below). The simulation

of LGM climate conditions follows the PMIP3 protocol (Braconnot et al., 2011, Braconnot et al., 2012). The concentrations of atmospheric greenhouse gases were lowered to their LGM values derived from ice core data (185 ppm for $CO_2$, 350 ppb for $CH_4$ and 200 ppb for $N_2O$) and the ice sheets are prescribed according to the product developed for PMIP3 (Abe-Ouchi et al., 2015). The model is run for several hundred years until the response to the LGM forcing in terms of surface climate variables is stabilised (Kageyama et al., 2013a, b). For this research, we extracted 50 years of monthly mean data (temperature 2m above

the surface, precipitation, sea level pressure and relative humidity) from the stabilised part of the simulation. Next, we downscaled the data, calculated their average climatology and their temporal (interannual) variability.

### 2.2 Generalized additive models

Generalized additive models (GAM, Hastie and Tibshirani, 1990) are statistical models blending the properties of generalized linear models with additive models. Given a dependent variable $Y$ and $p$ predictor variables $[X_1,\ldots,X_p]$, GAMs compute $\mathrm{E}(Y|X_1,\ldots,X_p)$, the expected value of $Y$, conditionally on the $p$ predictors $X_i$, as a sum of non-parametric functions as follows:

$$5 \quad \mathrm{E}(Y|X_1,\ldots,X_p) = \sum_{i=1}^{p} f_i(X_i), \tag{1}$$

Following Vrac et al. (2007), cubic spline functions were used for the $f_i$, represented by piece-wise third-order polynomial functions. For each function $f_i$, a number of knots are placed evenly throughout the predictor range, and the cubic polynomials that compose $f_i$ are constrained to continuity conditions at each knot to ensure smooth transitions (Wood, 2000, 2004). GAMs were calibrated using the mgcv package (Wood, 2011) in R, and the number of knots was determined automatically using generalised cross-validation.

Using a combination of geographical and physical predictor variables has been shown to improve spatial downscaling results (Vrac et al., 2007). The GAM uses these predictor variables, in addition to the original variables to be downscaled, to correct the biases of the GCM described above. This implies that a given GAM is only valid for the GCM for which it was calibrated, as it corrects its' specific biases. Using GAMs on climate variables requires the predictor and dependent variables to have the same spatial scale. The present-day dependent variables (precipitation and temperature) are at a fine spatial scale. The elevation variable is also at a fine spatial scale, whereas the predictor climate variables generated by the GCM are at a coarser scale. Thus, interpolation of the predictor climate variables is necessary. For this research we tested three interpolation techniques (see below).

Once the functions $f_i$ have been fitted using the present-day data, the downscaling can be performed on the GCM outputs for the LGM. The downscaling uses fine-scale and interpolated predictor climate variables corresponding to the LGM to generate fine-scale dependent variables. Here, two GAMs are calibrated: one for temperature and one for precipitation.

**2.3 Calibration data**

**2.3.1 Fine-scale dependent variables: the CRU climatology.**

Fine-scale, present-day temperature and precipitation dependent data were obtained from the Climate Research Unit (CRU, New et al., 2002). The spatial resolution of the data is 10' (*i.e.* 1/6 degree), regularly gridded between 32.72° and 59.861° latitude (N = 164 values) and -11.578° and 24.738° longitude (N = 219 values) for a total of N = 35916 grid-points. We computed a monthly climatology for each gridpoint by averaging the variable of interest (temperature or precipitation) over 30 years (from 1961 to 1990) for each month (Fig. S1), resulting in 12 values for each cell. The GAM is calibrated over this 30-year climatology, because the GCM cannot be set up to generate predictor variables for a specific year.

We are specifically interested in downscaling temperature and precipitation over Western Europe, south of the ice-sheets during the LGM, a time when archaeological data indicates that human populations contracted in size and range. The GAMs were calibrated over a wider area than the region of interest in order to avoid edge effects and include the full range of climate conditions that prevailed during the LGM, which was much colder than the present day. As a result, the calibration domain extends further North and East (where more continental conditions prevail) than the region of interest. Preliminary simulations nonetheless showed that selecting too large a calibration region averaged-out the small-scale variation we are interested in.

### 2.3.2 Large-scale predictor variables.

We used the data from a CMIP5 historical simulation run with the IPSL-CM5A-LR model to produce the predictor climate variables used for the calibration of the GAMs. We calculated monthly climatological averages from the simulation outputs for the period from 1961 to 1990, i.e. the same years as the CRU data (see above). The predictor variables we used are: temperature (T), precipitation (P), atmospheric pressure at sea level (SLP) and relative humidity (RH). The variables were spatially interpolated to match the spatial resolution of CRU data, which is 10'; each grid-point in the CRU data therefore matches a value for each of the predictor variables. Three interpolation methods were tested: bilinear, bicubic and kriging (Figs. S2-S4). For the kriging method, we used the "krig" function from the vacumm python package (http://relay.actimar.fr/~raynaud/vacumm/) using an exponential fit of the variogram, with the fit computed independently for every month and every variable. Different interpolation methods generate differences in the fine-scale predictor data. For example, the bicubic interpolation generates values outside of the initial range of values, contrary to the other two techniques. The bilinear interpolation generates abrupt changes in the slope of the values at the initial data points, whereas the other two techniques generate smooth surfaces. It is therefore important to assess the potential impact of the interpolation method on the output of the downscaling process.

### 2.3.3 Fine-scale predictor variables.

We extracted present-day elevation data from the CRU dataset gridded at the same fine-scale spatial resolution as the dependent variables. We computed the advective (Aco) and diffusive (Dco) continentalities, following Vrac et al. (2007). Dco is bounded between 0 and 1, and corresponds to the shortest distance to the ocean. A low value means that distance to the ocean is small, and *vice versa*. Aco is also bounded between 0 and 1, and takes the direction and intensity of prevailing winds into account along with the distance to the ocean. The change of Aco during a time *dt* is governed by Equation 2:

$$dAco = [-Aco(1 - i_{co}) + (1 - Aco)i_{co}] \frac{U}{\frac{l_0}{U_0}} \ln 2 \ dx \qquad (2)$$

where $i_{co}$ is 0 over sea and 1 over land, *dx* is the distance traveled by the air mass during *dt*, *U* is the mean wind norm, obtained from the GCM, and $l_0/U_0$ is the distance/wind ratio corresponding to a change of Aco of 1/2. Both variables are used to account for the fact that an air mass becomes more continental as it travels across land. Since Dco and the Aco proved to be highly correlated but Aco provided the best performance in the models, we only selected Aco for the present analysis (Figs. S2-S4).

## 2.4 Calibration of the GAMs

For each dependent variable (temperature and precipitation) and for each interpolation technique (bilinear, bicubic and kriging), we tested different combinations of physical and geographical predictor variables. To downscale temperature, we computed GAMs for all possible combinations of coarse-grain temperature values from the GCM interpolated at fine scale, with fine-grain elevation and advective continentality (Aco), resulting in seven possible models for each interpolation. To downscale precipitation, we computed GAMs for all possible combinations of coarse-grain temperature, coarse-grain precipitation, sea-level pressure, and relative humidity values from the GCM interpolated at fine scale, with fine-grain elevation and advective continentality (Aco), resulting in 31 possible models for each interpolation. For each interpolation technique, the resulting GAMs were compared using the Akaike Information Criterion (AIC; Akaike, 1974), and the model with the lowest AIC was selected. The AIC is a measure of the relative goodness of fit of each of the models and penalizes the number of parameters, thus preventing overfitting. The significance of each variable was assessed using p-values, and verified by visual inspection of the spline 95% confidence intervals. Six GAMs were therefore retained after calibration (one for each response variable and for each interpolation technique; Table S1).

## 2.5 Downscaling of temperature and precipitation time series for the LGM

We computed downscaled temperature and precipitation values using the six GAMs resulting from the calibration process described above, i.e. for the same predictor variables. The large-scale climate variables were generated by the GCM using the PMIP3 protocol for the LGM prior to interpolation (Figs. S5-S7). The geographical variables are derived from a digital elevation model for the LGM (Levavasseur et al., 2011). In particular, the change in coastlines due to the lower sea-level at LGM is accounted for, which has an impact on the continentalities. The downscaling was performed for each month of the 50-year-long monthly output from the GCM, in order to obtain a long time series of fine-scale temperature and precipitation values over Europe and calculate climatological averages. We also calculated indices of variability, including measures of variance and inter-annual variability for the variables of interest. The standard deviation of monthly mean temperatures for each month was calculated for the 50-year run. The coefficient of variation of monthly mean precipitation values was calculated for the same period.

## 2.6 Evaluation data (palynological data and vertebrate remains)

To evaluate the performance of the SDM for the LGM we compared our temperature and precipitation outputs to local climate variables estimated on the basis of pollen and vertebrate fossils from 29 test locations (Table S1, Fig. 1). For each of our 29 test sites, we estimated the mean, minimum, and maximum temperature and precipitation rate on a monthly basis over the course of the 50 downscaled years, and compared the ranges of downscaled values to the ranges of temperature and
precipitation values reconstructed using the palynological data and vertebrate remains.

Reconstruction of local temperature and precipitation values (annual mean temperature, mean temperature of the coldest month, mean temperature of the warmest month, mean annual precipitation, precipitation in January, precipitation in July) were obtained from pollen data reported in an independent study using inverse vegetation modelling for 14 sites located in
Europe (Wu et al., 2007). For the remaining 19 sites, vertebrate remains from another study (Burke et al. 2014) were used to calculate bio-climate indices (BCI; ff. Hernandez Fernandez, 2001a, b). The method set forth by Hernandez-Fernandez uses large and small vertebrates to compute the relative probability that a given assemblage reflects one of Walter's nine global zonobiomes (Walter and Box, 1976). The method is based on the "climate envelope" method commonly employed in biogeographical reconstructions. The BCI uses presence/absence data, thus avoiding the problems inherent with calculating
the relative representation of species from the archaeozoological record, and all available taxa rather than one or two "indicator" species, thus avoiding the risk that changes in the distribution of a single taxon could bias the biogeographical reconstruction. Ranges of temperature and precipitation values (minimum, mean and maximum) for each zonobiome were estimated by mapping the modern distribution of zonobiomes in the northern hemisphere and compiling present-day temperature and precipitation data from the CRU data (see Burke et al. 2014). The zonobiomes (calculated using BCI) were
then used to predict the climate ranges for each test location. Note that the range of values for each zonobiome corresponds to the minimum and maximum values for the zonobiome over Western Europe (see Burke et al. 2014). It is therefore not specific to a given location and encompasses a large range of values. The intervals generated by these two different climate reconstruction methods, therefore, are not equivalent. Nevertheless, they provide useful references for evaluating the values generated by the SDM.

**2.7 Validation using present-day data (CRU 1950-1960 time series)**

Due to computational constraints, the validation of the SDM was performed for the kriging interpolation technique only, based on the comparison of the results of the downscaling for the LGM between the three interpolation techniques (see below). A GCM simulation was produced for the period from 1950-1960 and compared with a time series of temperature and precipitation at 10' resolution over Europe for the same period (Mitchell et al. 2004). Yearly averages and variability indices (standard
deviation for temperature and coefficient of variation for precipitation) for the two sets of data were compared. The time series was based on the same original data used to create the 1961-1990 climatology, which forms the calibration set for the SDM (New et al. 1999). The time series was created by spatially interpolating data from irregularly spaced climate stations using

thin-plate smoothing splines, however, and may be subject to its own bias. Potential differences between our results and the validation time series should therefore be interpreted with caution.

## 3. Results

### 3.1 The GAM

The best models (i.e., the models with the lowest AIC value) for temperature and for precipitation were obtained by using the same sets of variables (one for temperature, one for precipitation) in the GAM for the three interpolation techniques. The predictors for temperature are: simulated temperature from the GCM, elevation and advective continentality (explaining 95.80%, 95.29% and 95.63% of the variance for the bilinear, bicubic, and kriging interpolations, respectively). For precipitation, the predictors are: simulated temperature, precipitation, sea-level pressure and relative humidity from the GCM,

elevation, and advective continentality (explaining 64.65%, 64.79% and 65.43% of the variance for the bilinear, bicubic, and kriging interpolations) (Table 1). The p-values for all variables for all models were < 0.001.

The splines resulting from the calibration process for the downscaling of temperature values show that fine-scale temperature readings are related to the GCM temperature and to elevation in a linear fashion, and the differences between the three

interpolation techniques tested are negligible (Fig. 2). The fine-scale temperature is proportional to the GCM temperature but it is inversely proportional to elevation, which means that the coarse-grain temperatures generated by the GCM are higher than fine grain observations in regions of high elevation. This is expected because temperature generally decreases with increasing altitude and because in the coarse grain GCM, it is the average elevation over the grid box that is considered. Although the model including all three predictor variables produced the lowest AIC, advective continentality has a very limited impact on

temperature, as the values of $f$(Aco) remain close to 0. When applied outside the range of values for which they are calibrated, GAMs use a linear extrapolation of the splines. The range of values for elevation is similar for the present-day and the LGM (Fig. 3). Because of the increased land mass during the LGM (which correlates with a low sea-stand), there are more high values for advective continentality, but this difference has a small impact since the spline is relatively flat for this variable. As expected, temperature is lower during the LGM than for present-day. This has a limited impact on the projections because the

linear interpolation of the spline outside of the range of values used for calibration is consistent with the fact that the spline is relatively linear for temperature  and the few remaining values are within 10 degrees of the minimum temperature. For very low temperatures during the LGM, however, the SDM outputs should be interpreted carefully, as discussed below.

The splines for the downscaling of precipitation, in contrast, are non-linear (Fig. 4). The splines showing the influence of

temperature on expected precipitation rates show larger variations due to a low expected precipitation for both low and high temperatures, but high expected precipitation for middle-range temperatures. Although the expected precipitation increases monotonically with the interpolated precipitation rates, the spline values are lower than the GCM precipitation values and the

relationship is non-linear. The expected precipitation increases more rapidly for low than for high interpolated precipitation, in keeping with previous observations that GCMs (and even RCMs) overestimate drizzles, which may explain this correction (e.g. Gutowski et al.,2013). The three interpolation techniques produced similar splines for all variables, although the splines of the bicubic interpolation are slightly distinct from the other two interpolation techniques. The main difference is observed

for the spline of the bicubic relative humidity, which indicates lower precipitation for low relative humidity than the other two interpolation techniques. Differences between the splines of the bicubic interpolation and the other two techniques were expected, since this interpolation generates the most divergent values (Figures S8, S11). As a GAM will adjust the splines to compensate for the potential biases of a GCM, it will also do so to compensate for the specificity of an interpolation technique.

Advective continentality is the variable with the least impact on precipitation rates. The ranges of values for simulated precipitation, relative humidity, and for elevation are similar for the present-day and the LGM periods, and the distributions of the variable substantially overlap, indicating that the splines calibrated over the present-day period can apply for the LGM (Fig. 3). As for temperature, the spline of advective continentality is relatively flat, and the difference of range of values will have limited impact on the projections. The spline for the simulated atmospheric pressure at sea level has a positive slope for

high values. This spline does not represent a causal relationship, but simply indicates that the GCM tends to underestimate precipitation at high atmospheric pressure. The simulated atmospheric pressure at sea level is also higher for the LGM than for the present-day period. Nonetheless, since the atmospheric pressure is mostly lower than 1045 hPa during the LGM, and given the low slope of the spline on the right-hand extremity, this discrepancy should have little impact on the results. For temperature, the splines are relatively linear near the lower end of the range of present-day values, and the linear interpolation

of the spline at lower values for the LGM is therefore sensible.

### 3.2 Results for the LGM

### 3.2.1 Temperature

Downscaled annual mean temperature was very similar for the three interpolation techniques tested (Fig. 5). This was expected, since the splines for the GCM temperature and elevation for all three techniques are also very similar. Temperatures

interpolated with the bilinear and kriging techniques were more similar to each other than to the temperature interpolated using the bicubic technique before (Fig. S8) and after (Fig. 5) applying the GAMs. The differences between the bilinear and kriging techniques show a pattern corresponding to the original coarse-grain cells from the GCM. This illustrates the difference between the two interpolation techniques: kriging generates smoother variations than the bilinear interpolation, which generates discontinuous variations at the original points. This difference remained after applying the GAMs, showing the

impact of the interpolation technique used on the final outcome of the downscaling.

The main differences between the interpolated and downscaled temperatures occur in the northeast of Europe (Fig. S9), where downscaled temperatures are higher, especially in winter. This difference was also observed when comparing present-day CRU data with the interpolated GCM data (Fig. S10), although with a much lower amplitude. Northeast Europe is the coldest region of the study area for both the LGM and the present-day (Figs. S1-S7). Since the spline for temperature has a slope lower than 1 for low temperatures (Fig. 2), the GAM generates higher temperatures than the interpolated values, especially for very low temperatures which fall outside of the present-day range of values due to the linear interpolation of the spline. As a result, the difference between interpolated and downscaled temperatures during the LGM is lower in summer. The SDM also takes fine scale variations in topography into account, such as abrupt elevation changes in the Alps and Pyrenees (Fig. S9).

The range of temperatures for the 19 sites for which the BCIs were computed are in accordance with the temperature reconstructions, irrespective of the interpolation technique used (Fig. 6). Simulated temperature ranges fall within the reconstructed ranges corresponding to the BCIs and are within the reconstructed ranges from Wu et al. (2007) for all test sites, as shown by the overlap of the red and blue error bars with the diagonal (Fig. 6a,c,e). As noted above, since the BCI reconstruction considers the minimum, mean and maximum values for each zonobiome over the whole of Western Europe (see Burke et al. 2014), some downscaled temperature values may differ from the mean, but as long as the error bars overlap with the diagonal are still in accordance with the BCI climate reconstruction.

### 3.2.2 Precipitation

The three interpolation techniques tested here produce similar distributions of precipitation rates but, compared to the two other interpolation techniques, the bicubic interpolation produced the most divergent results (Fig. S11). All three interpolation techniques nonetheless reflect the biases of the GCM (Fig. S12). Precipitation rates predicted using bicubic interpolation were substantially lower than those predicted using the other techniques in the South-West of Europe in winter and higher in the North of Europe (over the current North Sea) in summer (Fig. 7). This result is consistent with the observation that the splines for the bicubic interpolation differed from the other two interpolation techniques. Despite a general agreement between simulated and observed annual precipitation mean over Europe (Dufresne et al. 2013), comparing GCM projections interpolated at fine scale with the present-day CRU data shows that the GCM overestimates precipitation over the South of Europe in winter and underestimates them over the North in summer (Fig. S13), which results in the non-linear spline for precipitation (Fig. 4). Coarse-grain GCMs are known to perform poorly when simulating the small-scale physical processes that drive local surface variables such as precipitation (Wood et al., 2004). This explains the discrepancies between the present-day simulations and the CRU data and, by extension, explains the adjustments performed by the SDM.

The precipitation ranges for the 19 sites for which the BCIs were computed are in accordance with the precipitation ranges reconstructed for the LGM. Simulated precipitation ranges for all sites fall within the reconstructed ranges corresponding to the BCIs, as shown by the overlap of the horizontal error bars (in red) with the diagonal (Fig. 8). The SDM predicts higher

precipitation values for 1 site (in North-West Iberia) than the reconstructions provided by Wu et al. (2007) (the simulated mean precipitation of the driest month was predicted to be higher than the maximum precipitation found by Wu et al.). However, while the BCI ranges correspond to minimum and maximum values over a relatively large spatial extent, the reconstructions offered by Wu et al. (2007) are site-specific and therefore produce smaller ranges of values. Moreover, Wu et al. (2007) based

their reconstructions on local adjustments of the biome estimates from the BIOME4 model (Kaplan et al. 2003). They therefore used the same initial values for different sites, which may underestimate differences between sites and explain the lower range of precipitation values compared to the values generated by the SDM.

### 3.2.3 Variability

The temporal variability of temperature and precipitation rates highlights differences between the three interpolation

techniques. When temporal variation was computed over the interpolated variables (Figs. S14, S15), bilinear interpolation displays regular spatial patterns for both temperature and precipitation, especially in summer. This pattern was less apparent for kriging, and almost absent for bicubic interpolation.

With bilinear interpolation, the interpolated values will necessarily be less variable (whatever the index used) than the original

values. For a simple linear interpolation, given 2 spatially consecutive values at 2 different points in time ($y(x_0,t_0)$, $y(x_0,t_1)$, $y(x_1,t_0)$ and $y(x_1,t_1)$), any linearly interpolated value $y(x_i)$ for a location $x_i$ in $[x_0, x_1]$ will necessarily be comprised in $[y(x_0),y(x_1)]$. In addition, due to the linear interpolation $|y(x_i,t_1)- y(x_i,t_0)| < |y(x_0,t_1)- y(x_0,t_0)|$ and $|y(x_i,t_1)- y(x_i,t_0)| < |y(x_1,t_1)- y(x_1,t_0)|$. By contrast, since kriging does not impose linear interpolation between $x_0$ and $x_1$, this relation does not necessarily hold, and even less for bicubic interpolation, which do not impose $y(x_0) \leq y(x_1) \leq y(x_1)$. However, because of this absence of restriction, bicubic

interpolation can generate values with a high variability, as for precipitation in summer in the South-West of the Iberian peninsula (Fig. S15; the high variability for kriging in winter only occurs at the boundary of the study area due to boundary conditions, and can therefore be discarded).

Computing temporal variation over the downscaled variables (Figs. 9, 10), showed that the GAMs attenuated this spatial

pattern, which nonetheless still occurred for the bilinear interpolation, and was almost non-existent for the other two interpolation techniques.

### 3.3 Validation on present-day data

The comparison of average downscaled (based on kriging interpolation) and observed (CRU) monthly temperature and daily precipitation for the 1950-1960 period shows that they are in good agreement (Figures S16, S17). For temperature, the main

difference occurs in the far North of the study area (Southern point of the Scandinavian peninsula), and on the Italian side of the Alps. The downscaled temperature is similar on both sides of the Alps, whereas there is some difference in the CRU data

(Figure S16), suggesting the inclusion of orographic wind as a predictor may improve the SDM for specific areas. For precipitation, the main difference occurs in areas of high precipitation, especially the Western coast of Great Britain (Figure S17). Nonetheless, these areas were the areas with higher levels of precipitation for both datasets.

The overall patterns of variability were overall similar between the downscaled and CRU datasets (Figures S18, S19), with nonetheless some local differences. Temperature variability was higher in the North-East region of the study area and lower in the South-West region, especially in winter (Figure S18), and the range of values for the standard deviation were very similar for the different seasons. Downscaling tended to slightly overestimate temperature variability in the North of the study area, and underestimate it in the North-East compared to the CRU data. Some small-scale differences are nonetheless difficult to

interpret, since the CRU data showed some spatial artifact, for example in the center of the Iberia peninsula. The amplitude of the variability values was more different for precipitation, with the variability observed in the CRU data being higher (Figure S19). Nonetheless, the general patterns were quite similar, with the Southern region of the study area having higher precipitation variability than the North for both the downscaled and the CRU datasets.

**4 Conclusion and Discussion**

We downscaled temperature and precipitation values produced by the IPSL-CM5 model for 50 simulated years over Western Europe during the last glacial maximum using a GAM, a computationally efficient method for downscaling GCMs (Vrac et al. 2007). A single GAM was used for each dependent variable, calibrated over an average of 30 years of present-day data. Comparing the outputs of the SDM with two different climate reconstructions showed that this method generates results that fall within the computed confidence intervals for the variables of interest. This enabled us to compute indices of climate

variability for the LGM in Western Europe. In a separate study, we were then able to test a suite of environmental predictors and demonstrate that climate variability is a key factor governing the spatial distribution of prehistoric human populations during the LGM (Burke et al. 2014, 2017).

Downscaled time series for a present-day period (1950-1960) falling outside of the calibration period (1961-1990) were in

good agreement with an independent time series for both averaged values and measures of variability. Our study, therefore, demonstrates that the SDM, originally designed to downscale climatology data (averaged over several decades), can be applied to a time series thus allowing us to compute spatio-temporal patterns at a fine scale appropriate for studying the spatial dynamics of prehistoric human populations. SDMs must be carefully parameterised, however, including selecting the appropriate size of the area used for calibration.

Overall, the downscaled temperature and precipitation values produced by the SDM are in agreement with the climate reconstructions obtained from vertebrate remains and palynological data, with few exceptions (Figures 6, 8). The SDM results for the LGM differ from the interpolated data in the northeast of the study area, reflecting the adjustments made in the GAM

to counter biases inherent in the IPSL-CM5A-LR GCM used in this study. These discrepancies have little consequence in the present study, since this region was covered by ice-sheets during the LGM. It was included for calibration because it represented present-day climate conditions that were close to those of the Southern part of the study area during the LGM. This region was also included in the downscaling to illustrate the fact that, since it was colder during the LGM than any present-

5    day region of the study area, results for this region should be interpreted with caution.

The choice of interpolation technique used in the SDM also proved critical as it has a strong impact on the distribution of climate variability. We tested three different interpolation techniques. Since GAMs require the predictor and the dependent variables to have the same spatial grain, bilinear interpolation is commonly used to downscale the coarse-grain data generated

by GCMs (Vrac 2007). However, as this research demonstrates, bilinear interpolation generates non-smooth surfaces which may cause spatial artifacts in the final output. We tested two other non-linear interpolation techniques which generate smoother surfaces: bicubic interpolation and kriging. Bicubic interpolation generates values outside of the initial range of values (and therefore under- or overestimates the values) but is faster to apply than kriging. Kriging is more computationally demanding but offers the advantage of constraining the interpolated values within the range of initial values. The three interpolation

techniques produced different results for both temperature and precipitation during the LGM (Figs. S8, S11). After applying the GAM, these differences were especially important for precipitation values (Fig. 7). Because the GCM generated coarse grain temperature values for present-day conditions which are highly correlated with the CRU data, all three interpolation techniques produced similar linear splines and led to similar results for this variable. In the case of precipitation, however, bicubic interpolation predicts drier environments than the other two techniques by up to 2 mm/day. Since GCMs operate at

grains that are too coarse to accurately model small-scale physical processes driving local surface variables (Wood et al., 2004), the SDM for precipitation relies on more variables than are required to model temperature. The splines for these variables are non-linear (Figure 4), however, and may exacerbate the differences between the bicubic interpolation and the other two techniques. The cumulative impact of the interpolation and the GAM can therefore be non-negligible. This highlights the utility of the comparison presented in this research, especially for local phenomena such as precipitation.

The variability maps produced when using bilinear interpolation show the presence of a spatial artefact, in the form of a regular grid, for both temperature and precipitation (Figs. S14, S15). This artefact reflects the fact that bilinear interpolation generates lower variability between the points from which the interpolation is performed. Although slightly attenuated, this artefact remained after applying the GAMs (Figs. 9, 10). Prior to applying the GAMs, bicubic interpolation produced maps with the

smallest level of artifacts, kriging was intermediate and bilinear interpolation produced the highest level of artifacts. However, bicubic interpolation sometimes generated unrealistically high variability for precipitation (Fig. S15) while the artifacts generated by kriging decreased after applying the GAMs (Figs. 9, 10). We conclude that although more computationally demanding than the other two techniques, kriging represents a good compromise between computational complexity and accuracy. Contrary to bicubic interpolation, kriging generates values within the range of the values generated by the GCM and

generates variability indices with more realistic patterns than the bilinear interpolation. We therefore recommend using kriging for SDM applications based on the method presented here.

The IPSL-CM5A-LR GCM is known to predict lower temperatures than the values observed at high latitudes in winter (Dufresne et al., 2013). This bias was indeed observed when comparing the interpolated temperature with the CRU present-day data. As a result, the spline for temperature has a shallow slope at low temperatures (Fig. 2). The resulting correction applied by the GAM was emphasised for the LGM data generated by the GCM in winter in the North of Europe (Fig. S9), which lie outside of the range of present-day temperature and therefore relied on a linear interpolation of the spline. The large differences in temperature are therefore likely to be a combination of an underestimation of temperature by the GCM, and an over-correction of the very low temperature by the SDM. The spatial domain used to calibrate the GAM is larger than the domain of interest, namely Western Europe south of the ice sheets (the region occupied by human populations during the LGM) for reasons discussed above. These include the necessity of avoiding edge effects and including the full range of climate conditions likely to have occurred during the LGM. The observed over-correction lies on the periphery of the calibration region and is not within the study region this SDM was designed for.

The calibration area selected should therefore be large enough to encompass a representative range of climate conditions but should overlap the study region in order to account for potential relationships between climate and geographical variables specific to the region. However, through trial and error we found that using too large a calibration region averages out these relationships and therefore runs counter to the objectives of the downscaling, which is to represent fine scale spatial variation. Further research into the impact of the size of the calibration region on the SDM would be an interesting avenue to pursue. In addition, some results presented here are probably highly influenced by the specific GCM that was used. Especially, the variability in the SDM results is sontrgly influenced by the variability of the original GCM, in addition to the choice of the interpolation technique (Figure 9, S14, 10, S15). For temperature, for example, for a given interpolation technique, the downscaling adjusts the interpolated GCM temperature based on elevation, which is constant for a given location. The SDM will therefore not change variability compared with interpolation of the GCM. Using ensembles of models increases confidence in climate projections by enabling a better quantification of such uncertainty (Tebaldi & Knutti 2007). Although the outputs of ensembles of models may be challenging to interpret, this is another promising avenue for improving the application of the SDM method presented here that should be pursued in the future, especially for the computation of variability indices.

Our goal in this research has been to develop and test tools for the production of climate simulations at suitable spatial and temporal scales for investigating the mechanisms through which climate change and climate variability may have affected human populations in the past. Our aim is to help explain some of the broad evolutionary patterns visible in the archaeological record. Our results demonstrate the potential of GAMs for the production of climate simulations at a fine scale of resolution, both spatially and temporally, at relatively low computational cost. The resulting climate simulations can be used to test human

decision-making at regional and local scales useful for investigating the spatial distribution of prehistoric populations against a backdrop of inter and intra-annual climate variability (e.g., Burke et al. 2017; Burke et al. 2014).

*Code and data availability*. The code used for the downscaling and the input and output data are available at at https://figshare.com/s/1b952e47ff274cc0687e (DOI: 10.6084/m9.figshare.5487145).

*Author contributions*. GLa, AB, MK, MV and GLe conceived the study. GLa and GLe implemented the R code for the downscaling. MK, MV and GR ran the IPSL-CM5A-LR simulations and implemented the code for the interpolations. CD

implemented the code for the computation of the continentality. GL wrote the paper and all commented on it.

*Competing interests*. The authors declare that they have no conflict
of interest.

*Acknowledgements*. This research was supported by the *Fonds Québécois de Recherche Société et Culture - Programme de soutien aux équipes* (demande no.179537). Jérôme Servonnat is thanked for his input on the IPSL-CM5A-LR model biases.

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

**Table 1. Model selection among all possible combinations of variables for the temperature and the precipitations. Only the five models with the lowest AIC are presented. AIC scores, differences in AIC compared to the lowest scoring model ($\Delta_{AIC}$), and AIC weights ($w_{AIC} = \exp(-0.5 \times \Delta_{AIC}) \in [0,1]$, representing the relative likelihood of the models) are reported.**

| Candidate model | AIC | $\Delta_{AIC}$ | $w_{AIC}$ |
|---|---|---|---|
| Bilinear interpolation | | | |
| *Temperature* | | | |
| T+elv+Aco | 869468.4 | 0 | 1 |
| T+elv | 878862.9 | 9394.52 | 0 |
| T+Aco | 951357.5 | 81889.14 | 0 |
| T | 955986.9 | 86518.50 | 0 |
| elev+Aco | 1605682.9 | 736214.55 | 0 |
| *Precipitations* | | | |
| T+P+elev+Aco+RH+SLP | 275068.7 | 0 | 1 |
| T+P+elev+Aco+SLP | 281222.3 | 6153.56 | 0 |
| T+P+elev+Aco+RH | 283266.4 | 8197.70 | 0 |
| T+P+elev+RH+SLP | 288574.6 | 13505.85 | 0 |
| T+P+elev+Aco | 288864.1 | 13795.35 | 0 |
| | | | |
| Bicubic interpolation | | | |
| *Temperature* | | | |
| T+elv+Aco | 896517.0 | 0 | 1 |
| T+elv | 904724.6 | 8207.52 | 0 |
| T+Aco | 952109.7 | 55592.63 | 0 |
| T | 955079.5 | 58562.43 | 0 |
| elev+Aco | 1605402.4 | 708885.36 | 0 |
| *Precipitations* | | | |
| T+P+elev+Aco+RH+SLP | 274137.9 | 0 | 1 |
| T+P+elev+Aco+SLP | 278178.8 | 4040.881 | 0 |
| T+P+elev+Aco+RH | 283162.3 | 9024.412 | 0 |
| T+P+elev+Aco | 286857.5 | 12719.589 | 0 |
| T+P+elev+RH+SLP | 286926.3 | 12788.379 | 0 |
| | | | |
| Kriging | | | |
| *Temperature* | | | |
| T+elv+Aco | 878970.1 | 0 | 1 |
| T+elv | 888326.4 | 9356.24 | 0 |
| T+Aco | 952033.1 | 73062.92 | 0 |
| T | 956903.8 | 77933.70 | 0 |
| elev+Aco | 1607626.2 | 728656.08 | 0 |
| *Precipitations* | | | |
| T+P+elev+Aco+RH+SLP | 269767.4 | 0 | 1 |
| T+P+elev+Aco+SLP | 276171.0 | 6403.59 | 0 |
| T+P+elev+Aco+RH | 277692.0 | 7924.60 | 0 |
| T+P+elev+Aco | 283099.2 | 13331.79 | 0 |
| T+P+Aco+RH+SLP | 286495.6 | 16728.23 | 0 |

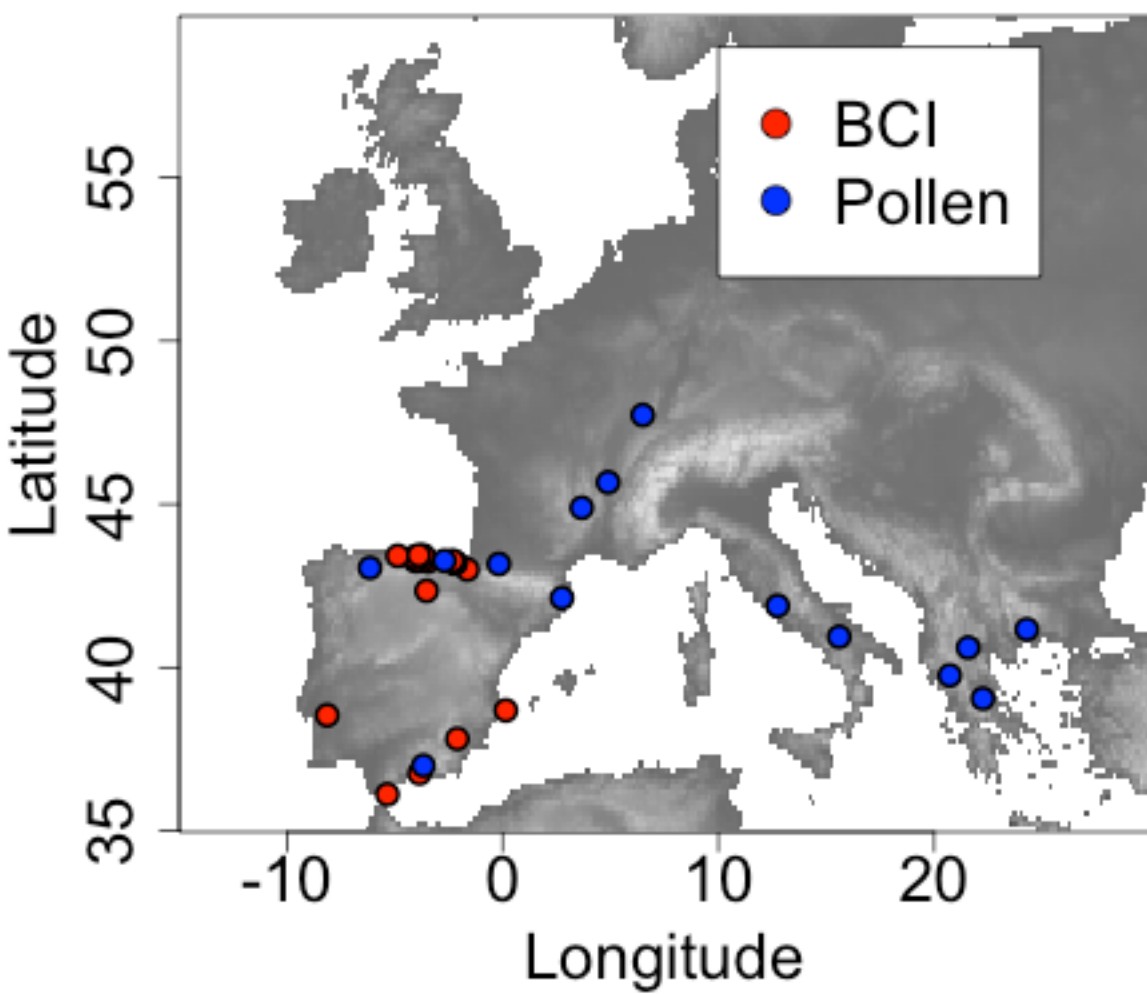

**Figure 1. Study area and locations of the sites used for reconstructing local climate variables estimated on the basis of pollen and vertebrate fossils, used for the evaluation of the method. The grey scale represents elevation.**

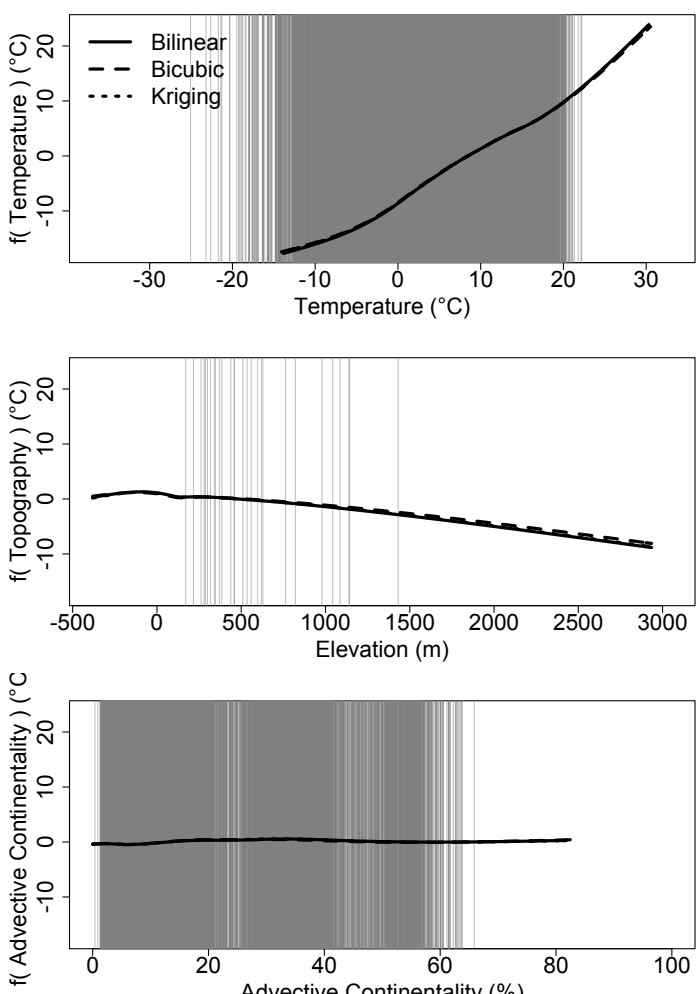

**Figure 2. Splines of the GAM for temperature. The splines are scaled to the same range to allow for visual estimation of their relative importance. The range of the x-axes combines the ranges of values for the present-day period and the LGM. The grey lines indicate the values for the 12 months over the 50 years during the LGM at the archaeological sites of Figure 1 (except for the elevation, for which there is only one value per site).**

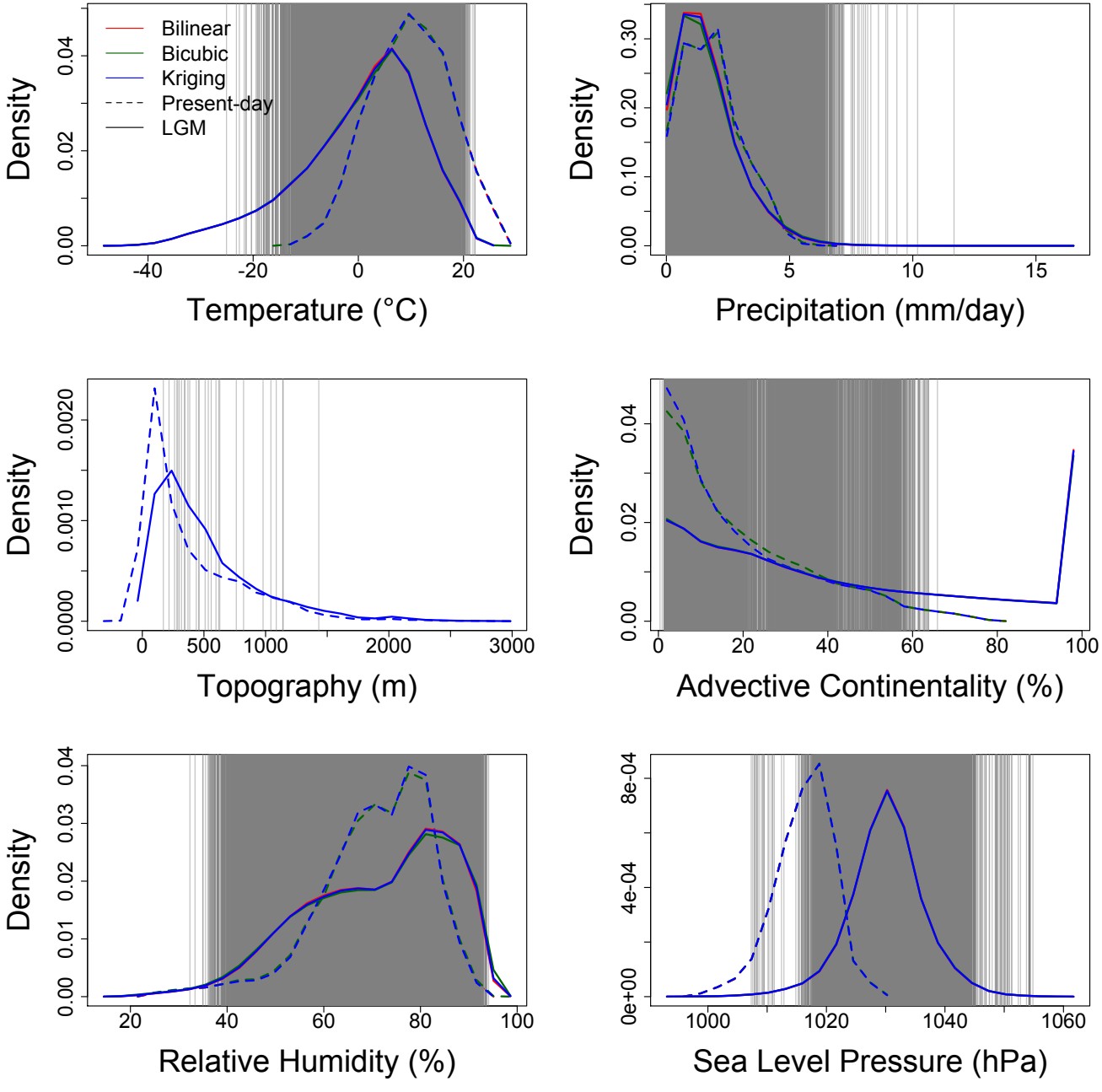

**Figure 3. Histograms of the predictor variables for the present-time (1961-1990; dashed lines) and for the LGM (solid lines) over Western Europe, using the bilinear (red), bicubic (green) and kriging (blue) interpolations. The grey lines indicate the values for the 12 months over the 50 years during the LGM at the archaeological sites of Figure 1 (except for the elevation, for which there is only one value per site).**

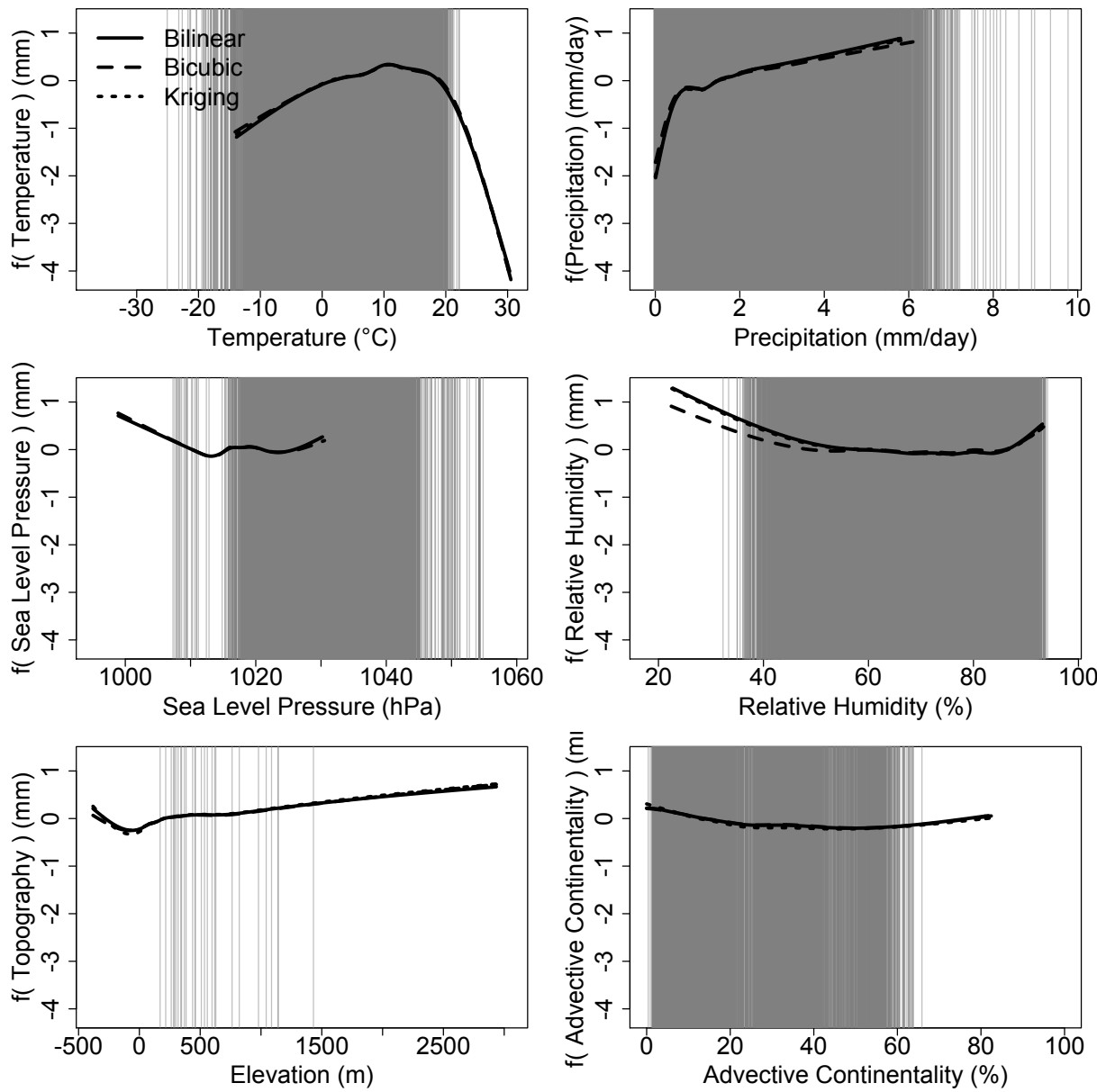

**Figure 4. Splines of the GAM for precipitations. The splines are scaled to the same range to allow for visual estimation of their relative importance. The range of the x-axes combines the ranges of values for the present-day period and the LGM. The grey lines indicate the values for the 12 months over the 50 years during the LGM at the archaeological sites of Figure 1 (except for the elevation, for which there is only one value per site).**

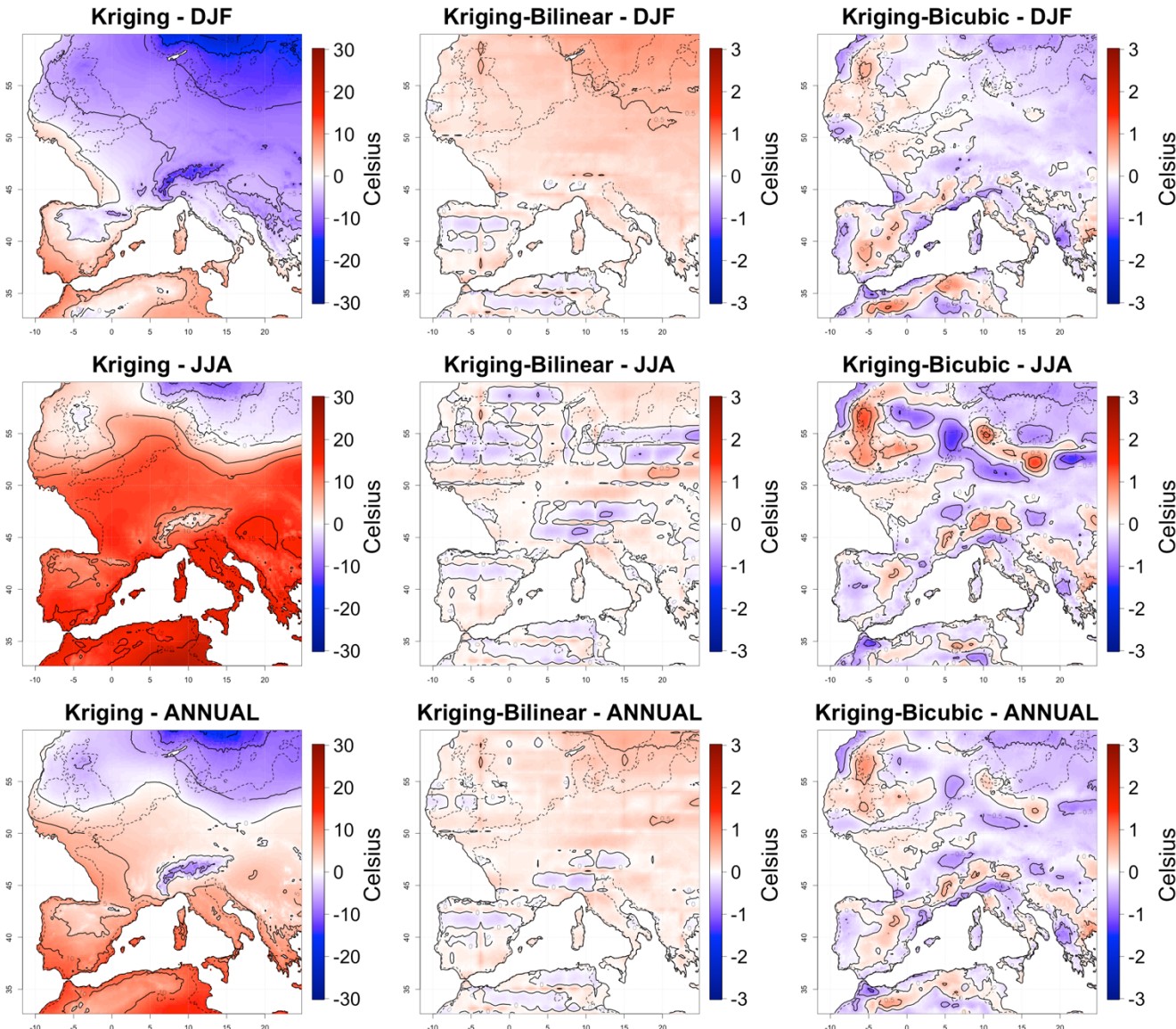

**Figure 5. Mean distributions of monthly mean downscaled temperatures over Western Europe during the LGM for winter (December, January, February), summer (June, July, August), and the whole year, computed over 50 years for the kriging interpolation technique, and difference between the kriging and the other two techniques. Downscaling was performed for each month independently, but results are combined into seasons to summarise the results.**

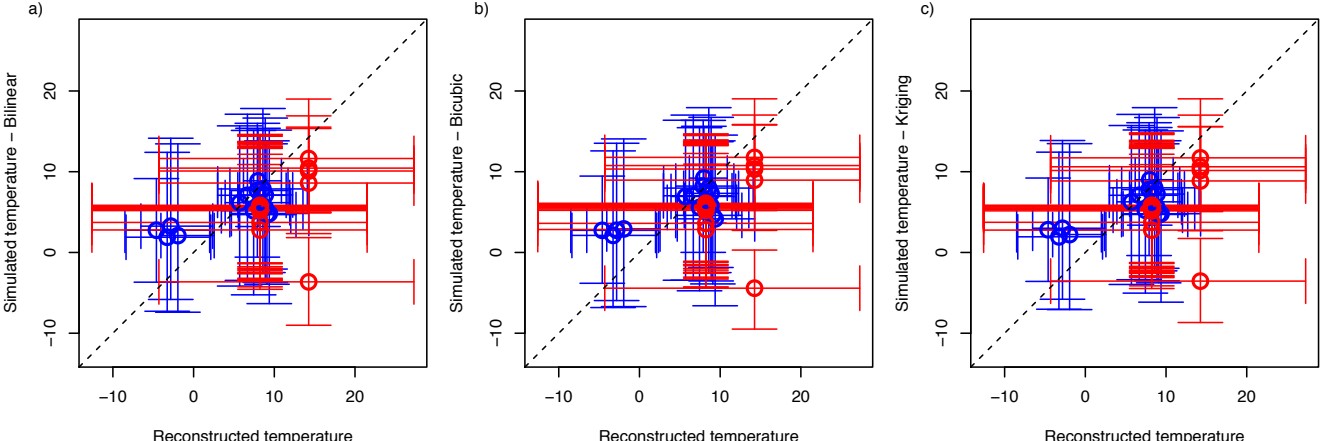

**Figure 6. Comparison of reconstructed vs. downscaled temperatures for the LGM based on the BCI indices (red), and from Wu et al. (2007)'s reconstructions (blue) for a) the bilinear, b) the bicubic and c) the kriging interpolations. The circles represent the mean temperature values for the two reconstruction methods (x-axis) and the downscaled values over the 50 simulated years (y-axis). The horizontal error bars represent the range of temperature values for the reconstruction method (minimum and maximum over the whole zonobiome for the BCI indices, mean temperature of the coldest and warmest month for Wu et al. 2007). The vertical error bars correspond to the mean temperature of the coldest and warmest month over the 50 simulated years.**

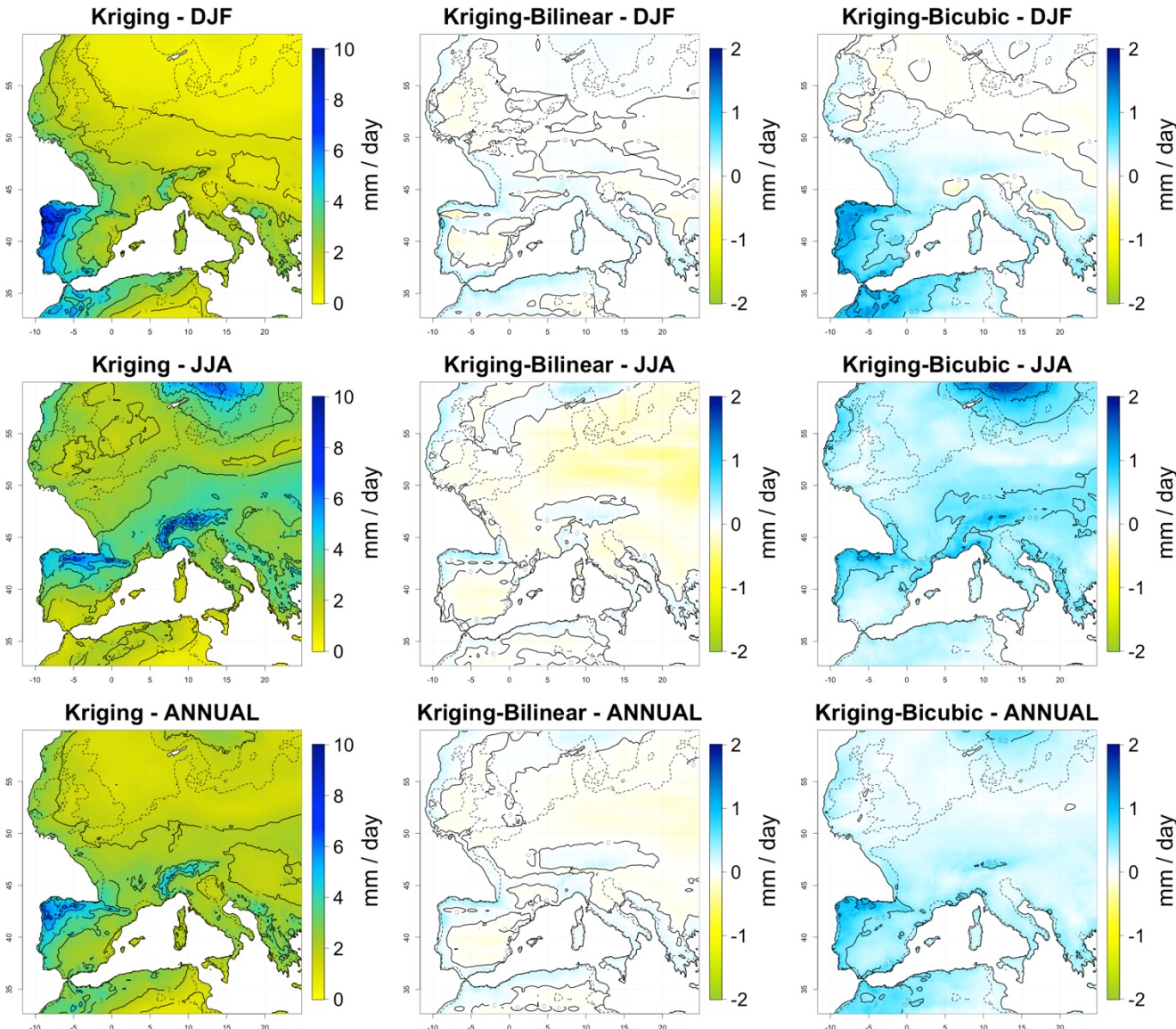

**Figure 7. Mean distributions of downscaled daily precipitations over Western Europe during the LGM for winter (December, January, February), summer (June, July, August), and the whole year, computed over 50 years for the kriging interpolation technique, and difference between the kriging and the other two techniques. Downscaling was performed for each month independently, but results are combined into seasons to summarise the results.**

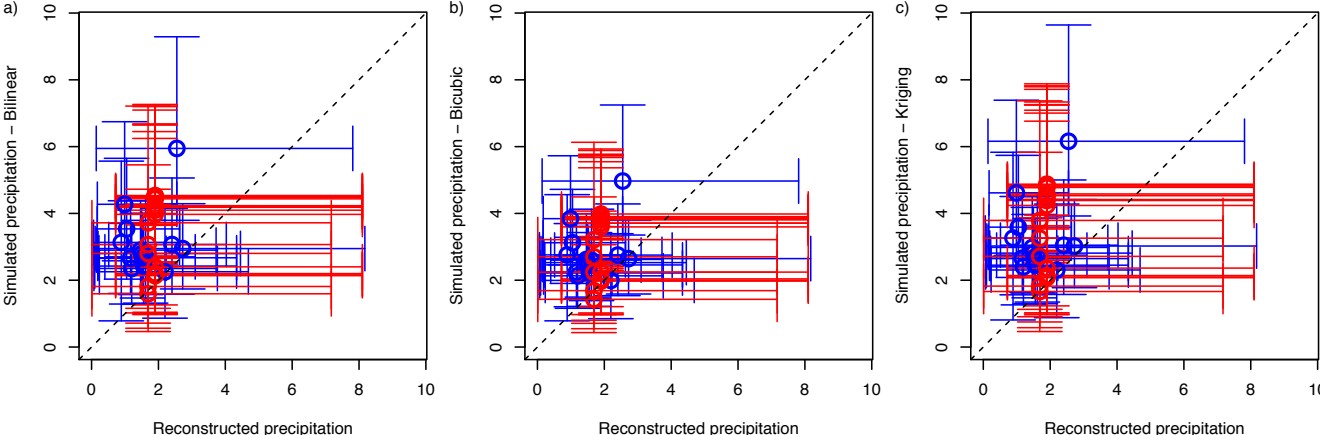

**Figure 8. Boxplot of reconstructed vs. downscaled precipitations for the LGM based on the BCI indices (red), and from Wu et al. (2007)'s reconstructions (blue) for a) the bilinear, b) the bicubic and c) the Kriging interpolations. The circles represent the mean temperature values for the two reconstruction methods (x-axis) and the downscaled values over the 50 simulated years (y-axis). The horizontal error bars represent the range of precipitation values for the reconstruction method (minimum and maximum over the whole zonobiome for the BCI indices, mean precipitation of the coldest and warmest month for Wu et al. 2007). The vertical error bars correspond to the mean precipitation of the driest and wettest month over the 50 simulated years.**

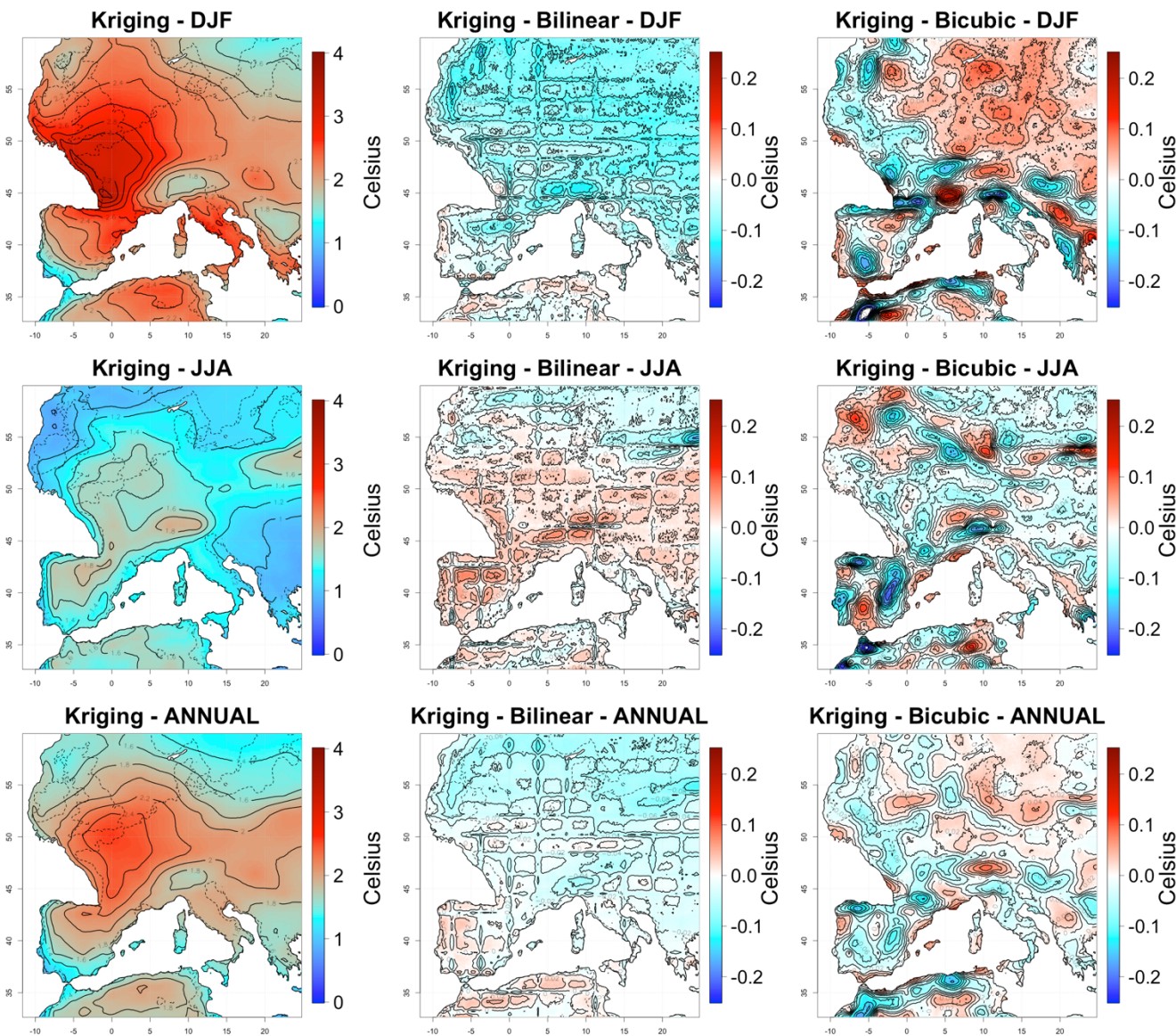

**Figure 9. Maps of temporal variations (standard deviation of each month across 50 years) of downscaled monthly mean temperatures over Western Europe during the LGM averaged over winter (December, January, February), summer (June, July, August), and the whole year for the kriging interpolation technique, and difference between the kriging and the other two techniques. Variability was computed for each month independently, but results are combined into seasons to summarise the results.**

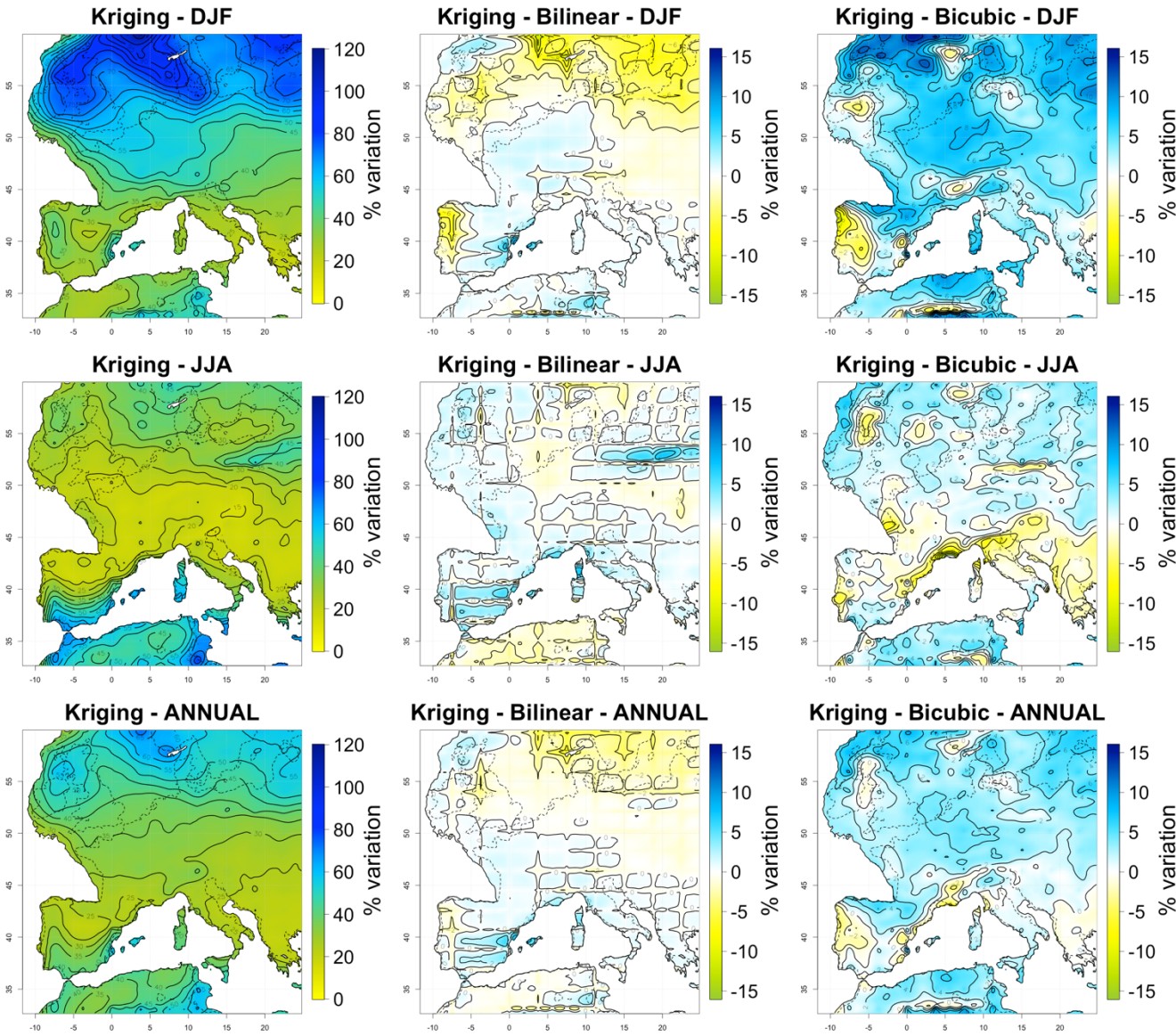

**Figure 10. Maps of temporal variations (coefficient of variation of each month across 50 years) of downscaled daily precipitations over Western Europe during the LGM averaged over winter (December, January, February), summer (June, July, August), and the whole year, computed over 50 years for the kriging interpolation technique, and difference between the kriging and the other two techniques. Variability was computed for each month independently, but results are combined into seasons to summarise the results.**