# Peer review of "Comparison of spatial downscaling methods of general circulation model results to study climate variability during the Last Glacial Maximum"

_Geoscientific Model Development, 2017_

## Referee Comment (RC1) · Anonymous Referee #1 · 6 Dec 2017

Review for: Comparison of spatial downscaling methods of general circulation models to study climate variability during the Last Glacial Maximum

This study applies a Generalized Additive Model to statistically downscale precipitation and temperature over Europe during the Last Glacial Maximum. It specifically evaluates the effect of different interpolation schemes (bilinear, bicubic and kriging) to the application of a previously used downscaling method (Vrac et al. 2007 as cited in the manuscript). I believe this manuscript could be accepted subject to revisions concerning the following issues.

The first issue involves the coarse scale GCM predictor variables used by the down-

scaling model. Using a single GCM (IPSL-CM5A-LR) to calibrate the model and generate simulations is problematic as it leaves the analysis subject to the biases of that individual model (biases identified in "European temperatures in CMIP5: origins of present-day biases and future uncertainties" for example). This GCM has a larger than average climate sensitivity relative to the CMIP5 ensemble and it's response to significantly reduced GHG concentrations may be similarly different from other GCMs. Calibrating a single GCM over a 30-year period should eliminate any biases due to inter-annual or decadal variability but could still influenced by lower frequency modes of variability. Magnitudes of temperature and precipitation in paleoclimate simulations could be amplified or diminished depending on whether the model was fitted in a generally cooler or warmer phase of low-frequency variability. Using an ensemble of models generally limits this effect as well. If only one GCM is feasible, then its characteristics and limitations should be explained in more detail.

The second concern is in the results for Generalized Additive Model (Section 3). There is confusion between the text and Figures 2 and 3 about what is occurring. On Page 8 starting with lines 34-35 and continued onto next page the text states: "Simulated atmospheric temperature at sea level was lower for the LGM than for the present-day period". Is this true? In Figure 2 the legend suggests present-day SLP is lower, while the caption suggests LGM SLP is lower (the figure legend and caption contradict each other over what the solid and dashed lines represent). Further, the domain of the spline for SLP (fitted in present-day) in Figure 3 is 1000 hPa to 1030 hPa which corresponds to the lower valued histogram in Figure 2, contradicting the text.

If the spline for SLP in Figure 3 is correct then it implies LGM SLP is described by the solid line and is higher than present-day SLP. This is (hopefully) correct because if the text and Figure 2 caption are correct, the temperature panel would imply that the LGM had high temperatures than present-day, suggesting there is something seriously wrong with the IPSL-CM5A-LR GCM! If the splines of Figure 3 are correct, then a linear extrapolation of the SLP spline into the higher SLP values of the LGM suggests

**[GMDD](https://www.geosci-model-dev-discuss.net/)**

Interactive
comment

precipitation will have a strong positive response to increasing SLP. This does not seem physically realistic.

The large differences in temperature in Figure S11 between downscaled and interpolated GCM values also raise doubts about the linear extrapolation of the splines to lower temperatures. The GAM is clearly adjusting the GCM temperatures upward in the majority of the region in response to what appears to be a cold bias in the GCM shown in S12 (the order of subtraction should be specified in both figure captions to confirm this). But does that mean in the LGM the GCM has a 20 degree C cold bias and the GAM is correcting this? Or is the GAM overcompensating and generating temperatures that are too warm because the slope of the temperature spline is too low?

A useful check of the downscaling model's performance would be to simulate the years in the historical model run (1901-1950 if available, 1951-1960,1990-2005) outside the calibration period and ask how the method performs against CRU observations before attempting to employ the method in time period with substantially different atmospheric forcing conditions. I suggest repeating the figures of S12 and S16 but comparing downscaled values (for winter, summer and using the different interpolation methods) against the CRU observations. If these figures replaced Figures 5 and 8 (moving those to the supplementary figures), it would provide a better picture of the method performance.

It may be beyond the scope of this study but it would be useful to see the GAM fitted separately using proxy data from the 29 sites in past and in present to see how the splines vary between such different climate regimes and whether linear extrapolation is indeed a good assumption.

The third concern is regarding the comparison of simulated temperature and precipitation against paleo-reconstructions during the LGM. The boxplots of Figures 6 and 9 do not clearly support the claim that the downscaling method is in "good agreement"

[Figure]

with the reconstructed values. I suggest removing (or moving to supplementary) the bilinear and bicubic panels and instead display comparisons of annual maximum, minimum and mean values separately for the kriging simulations using proper boxplots. This would provide a clearer comparison between the actual values and allow for at least a visual comparison of the distribution of these values over the 50-year LGM period to be compared. Additionally, how do you measure model performance when the two selected proxy biomes are significantly different from one another (as occurs more often for precipitation)?

It would also be particularly useful to evaluate the performance of the GAM in replicating present variability outside the calibration period given the importance of climate variability for human population distributions. Figures S19 illustrates the differences between interpolation methods in the LGM but doesn't show whether the GAM is simulating the variability accurately. It would be useful to see SPI and STI from the GAM compared to the same values for CRU similar to S12 and S16.

Further to Figure S19, maps showing the differences between the interpolation methods, as presented in the figures before, would help illustrate the effect of the different methods more clearly. Are the differences in variability from the three methods meaningful and if so are they large enough to suggest the methods could imply different patterns of human migration?

Specific/Technical Comments:

P1 Line 17: Remove the "s" from methods in "Statistical Downscaling Methods".

P1 Line 27: In the sentence beginning with "Our results" replace "confirming" with "suggesting", add "is" before "suitable" and drop "is sound" at the end. The current sentence is too strong given the evidence presented.

P1 Line 31: Replace "their" with "the".

P8 Line 3: I am skeptical that the p-value for ACO is so low (particularly for temperature)

given the sensitivity of the GAM to ACO is so small. If the variance explained by ACO is indeed statistically significant, the splines and the AIC values would suggest it is not meaningfully significant. This is noted in later paragraphs on this page.

P8 Line 8: The inverse proportionality of temperature to elevation in the GAM spline does not itself imply that the GCM overestimates temperature at high elevation (though it likely does for the reason stated in the next sentence). It merely implies, that in the GAM if elevation increases while the other parameters are constant, then the simulated temperature is expected to decrease.

P8 Line 16: Sentence beginning with "This should...". The curvature of the lower end of the temperature spline is not negligible so this is not necessarily a safe assumption.

P10 Line 6: Add "s" to "underestimate".

P10 Line 30-31: Given the boundary condition issue is present for all the interpolation methods, why not reduce the applicable study area to exclude the outer regions where the downscaled values will be unreliable?

P11 Line 15: "Satisfying results" is subjective, prefer a quantifiable description of how the results compared.

P11 Line 16-16: Sentence beginning with "Elsewhere" seems misplaced here.

P11 Line 20: "Critical" is too strong a descriptor here. This study shows the choice of interpolation can reduce spatial artifacts but does not explicitly demonstrate that it alone is most responsible for the GAM accuracy.

P11 Line 31-32: "non-linear" One could have linear splines and still end up with differences due to choice of interpolation method.

P12 Line 12: "Assuming ... accurate". This is not a good assumption and I suggest simply starting the sentence at "We conclude".

P12 Line 17: "Reliable temperatures"? There are significant biases in the mountains

as shown in Figure S12.

P12 Line 21 starting with "This correction" to the end of the paragraph: Isn't this further evidence that the domain of the study area should be reduced to areas with paleo proxy data and without coverage by an ice sheet?

Figure 1 and 3: Add the linearly extrapolated splines in a different colour to show how the variables would respond in regimes that occur during the LGM.

Figure 2: Correct the labelling contradiction between the legend and the caption.

Figure 3: Are the units for the precipitation spline "mm" or "mm/day"?

Figures 6 and 9: Please revise the y-axis ranges of the boxplot figures to span the actual range of data displayed (e.g. there are not any temperature values above 40C yet the plot extends beyond 60C).

Figures 10 and 11: I understand the colour scales here vary from panel to panel to highlight spatial artifacts but it makes interpreting the relative effects of the methods more difficult. I think common colour scales would be more useful given the spatial artifacts should be visible from the contours anyway.

Figures S1 through S7: There are too many individual panels within these figures and they have insufficient resolution which makes them impossible to read. I suggest presenting only the four seasons for S1, and a few representative panels from the different interpolation methods for S2 -S7 which would allow them to be presented at a readable scale.

Figures 10 and S17: Please revise the red-black colour schemes to something analogous to the other figures. The large magnitude darker colours obscure the contours and make large areas of the map seem overly homogeneous.

Figures S17 and S18: Add a note in the caption why a different land-sea mask is used in these figures relative to all of the others. I suspect it is because the Mediterranean
illustrates the differences in interpolation technique quite well. However, if these are masked out and not used for projections in the LGM it also raises the question of whether these differences are meaningful in the areas actually used in the analysis.

Figure S19: Specify this is during the LGM. "(STI and SPI values in ]-1,1[)" is a typo?

———————————————

---

## Referee Comment (RC2) · Anonymous Referee #2 · 22 Dec 2017

This manuscript deals with the application of a downscaling technique combining interpolation (through three techniques) and General Additive Models (GAMs), over Western Europe during the Last Glacial maximum (LGM). Results are compared to site-specific climate proxys from pollen and vertebrate remains data. Its seems well within the scope of Geoscientific Model Development, and deals with the relevant topic of developing statistical downscaling tools that may be used in very different climates like the LGM. The manuscript needs in my opinion some tightening of the objectives, some work on the clarity of the text and take-home messages, as well as some additional simulation analysis. I detail below these few main comments, together with many specific ones. I can therefore recommend publication of the manuscript only once all these

comments are addressed.

**Main comments**

1. It is not clear from the start (and down to the choice of figures)what are the objectives of this manuscript. Is it the comparison of downscaling methods (i.e. through different interpolation techniques)? Is it the adequate simulation of reconstructed climate proxy data? Is the target location the whole Europe or only the proxy specific sites? All these questions should be answered from the beginning of the manuscript. As they are currently not answered, the organization of the manuscript and the choice of figures are indecisive (see specific comments below).

2. The simulation set-up clearly lacks some present-day validation, as already pointed out by reviewer #1. This would hopefully help disentangling errors/biases from the interpolation, GAM models, and the driving GCM (see specific comments below).

3. Another consequence of the first main comment above is that a large number of supplementary figures are commented in the main text, which is quite frustrating for the reader. The organization of figures (and associated text) should definitely be redesigned (see specific comments below).

4. In relation to the second main comment above, there is little uncertainty discussed in the manuscript, be it a result of the short calibration period for GAMs or from another source like using a single GCM. This should definitely be taken on by the authors for the manuscript.

**Specific comments**

1. P3L1: Please define "taphonomic"

2. P3L23-25: The length of the two GCM simulations is not clear here

3. P4L29: The reference used here for the mgcv R package is not one of those recommended in the citation info of the package. Please correct this.

4. P5L1-2: Is there actually a theoretical reason for the requirement of the same scale? I fully understand the advantage of using e.g. downscaled precipitation as a predictor for local precipitation, but when considering other predictors like SLP, the most informative scale for local precipitation my clearly not be the local scale, but a larger domain shifted in the direction of the prevailing winds (at least in western Europe). This would open quite different approaches for performing this kind of studies that would not require the interpolation step. But this may lead to difficulties given the change in land/sea mask and the presence of ice caps when considering LGM simulations. I would appreciate a comment on that.

5. P5L13-15: This starts to be confusing in terms of data. I believe that (for any location) not only the monthly regime (i.e. 12 values only) is used, but the whole 31-year monthly time series. Please be more specific.

6. FigureS1: This figure is not readable at all. Same for Figures S2 to S7. I would strongly recommend finding a way to make them actually useful.

7. P528-P6L3: "Aco" should be defined mathematically in the text without having to look into Vrac et al. (2007). There is no need to define "Dco" if not used, apart maybe from writing that it is highly correlated to "Aco".

8. P6L16: There should be a reference here to Table S1.

9. Section 2.5: There should be two additional subsections on the interpolation/downscaling for the present-day reference period (1960-1990) and for a present-day validation period (see Main comment above and comments from reviewer 1).

10. P6L24: "Extremes" is a much too strong word here. This set-up (length of time series and temporal resolution) prevents assessing extremes.

11. Figure S8: This map should definitely be included in the main text, because this critically shows where to look in European map results. It might also be relevant to systematically indicate these locations in the results maps (depending on their size).

12. P7L18-25: Results on the SPI and STI are not used at all in the manuscript (only in the Supplementary material), so please remove their description (and possible comments) from the main text.

13. P7L18-25: The description of SPI computation lacks many important details: (1) what is the climatic norm, i.e. the reference period over which the standardization is based (present-day, LGM) and why? (2) what is the chosen distribution function for monthly precipitation? (3) Is it the same everywhere in Europe? Results are quite sensitive to these issues, as clearly shown in the literature (see e.g. Wu et al., 2005; Stagge et al., 2015)

14. P7L18-25: The choice of a variability index as the number of months with SPI between -1 and 1 is actually very strange (and indeed quite irrelevant). The SPI is by definition normally distributed, so the probability of having a SPI between -1 and 1 is 68.27%, which amounts to around 410 months in 50 years (95% confidence interval: 387-432), if the reference period for fitting the precipitation distribution is the same as the computation period (which I believe is the case here, see P7L20). So the spatial pattern observed in Figure S19 is a complete

artifact due to (1) the limited length of the period used for fitting the distribution, and (2) the relevance of the specific theoretical distribution used for fitting. Based on the 3 above comments, I strongly suggest removing all the analysis done with SPI/STI.

15. Figure 1: It would be great to see the range of present-day and LGM predictors in these figures in order to directly check statements made in the text P8L12-18.

16. Figure 2: Like reviewer #1, I believe that dotted lines are for the present-day period. Please remove the wrong legend definition from the caption.

17. Section 3.2: As mentioned above for section 2.5, there should be an additional result section for validation the interpolation/downscaling process in a present-day period distinct from the calibration period.

18. Most of results are presented at the annual time scale of for two 3-month seasons. What is then the advantage of fitting GAMs for individual months? I would expect a larger explained variance for annual or seasonal averages. I would appreciate some comments on this issue in the manuscript.

19. Figure 4: Possible differences between the three interpolation techniques cannot be appreciated from these maps with a common colour scale, because of (1) the large spatial range, and (2) the large seasonal range. Figure 5 looks into all possible differences between the 3 techniques, making both figures relatively redundant. I would therefore recommend choosing one interpolation technique as reference (ideally the one that should be recommended in the conclusion of the manuscript) and plot (1) maps as in Figure 4 for this technique, and (2) differences from this reference with a specific colour scale, as in Figure 5. This would hopefully reduce the number of figures and make the message clearer ("we choose this technique and results with the others are not that different.")

20. Figures S9 to S12. This is a much too high number of figures which shows that work on synthesizing results is clearly lacking. The reader should be presented two things: first, how temperature (and precipitation in a second step) is transformed by the whole downscaling process, through maps of raw, interpolated, interpolated +downscaled, and observations (CRU) in the present-day period. A similar presentation should be made for the validation period, and for the LGM period (for which CRU observations may be replaced by the pollen and vertebrate proxys). This could be made only for the reference interpolation technique. Second, additional maps should show the differences with the two other techniques, possibly through the whole downscaling process. This would require reorganizing figures and text (P9L6-28), but for a much better clarity of the manuscript!

21. P9L25-28, and Figure6: I am not convinced by results as presented here, as these plots are not very appropriate for identifying agreement for each site independently. I would therefore recommend trying scatterplots (with uncertainty bars as here or better uncertainty squares), with reconstructions (BCI, Wu et al. data) on the x-axis and simulations from this paper on the y-axis. The overlap of uncertainty ranges with the diagonal might better inform on the agreement of simulations with reconstructions.

22. P9L30-P10L16: cf. comments on temperature for an additional validation period, revised figure organization, etc.

23. P10L9-11, "This is due... such as precipitation (Wood et al., 2004)": I don't understand why this should lead to the European-scale discrepancies noted in the previous sentence. Please make it clearer.

24. P10L23-31: I find this paragraph a bit long, compared to other issues elsewhere that would also deserve some explanations.

25. P11L5-9: As mentioned above, please remove the SPI/STI analysis and results.

26. P11L25-26: So should we use kriging? Please be more specific on your conclusions about the preferred interpolation method.

27. P12L14-15, "more reliable variability": I don't understand. Please make it clearer.

28. P12L19-29: Well, this clearly poses the question on whether one should put confidence in GCM outputs at high latitudes (at least. . .). And for this study, this raises the following issue: should the interpolation/downscaling take place over the whole of Europe for reconstructing only a few sites located in the south of the continent. This issue should be seriously taken into account by the authors for the manuscript. Indeed, there may some biases in LGM results in the south due to present-day biases in the north via the continent-wide GAM modeling. . . I am definitely expecting comments on this potential issue.

29. P13L3, "larger-scale patterning": Could you explain and make it clearer?

30. P13L4-6: I am not sure this sentence is relevant here.

31. Table 1, "AIC weights": this should be defined and commented in the text.

**Technical corrections**

1. P5L25: Remove "interpolated variable"

2. P6L29-30: Redundancy of "downscaled, simulated"

3. P7L3: Please specify that "bio-climatic indices" is abbreviated as BCI(s) in the following.

4. P8L10: font size of "predictor"

5. P11L31: "than for the temperature"

**References**

Stagge, J. H., Tallaksen, L. M., Gudmundsson, L., Van Loon, A. F. Stahl, K. (2015) Candidate distributions for climatological Drought Indices (SPI and SPEI). International Journal of Climatology, 35, 4027-4040. doi: 10.1002/joc.4267

Wu, H., Hayes, M. J., Wilhite, D. A. Svoboda, M. D. (2005) The effect of the length of record on the standardized precipitation index calculation. International Journal of Climatology, 25, 505-520. doi: 10.1002/joc.1142

---

## Author Comment (AC1) · 1 Apr 2018

This study applies a Generalized Additive Model to statistically downscale precipitation and temperature over Europe during the Last Glacial Maximum. It specifically evaluates the effect of different interpolation schemes (bilinear, bicubic and kriging) to the application of a previously used downscaling method (Vrac et al. 2007 as cited in the manuscript). I believe this manuscript could be accepted subject to revisions concerning the following issues.
We thank the reviewer for taking the time to review our manuscript and for providing the constructive comments below.

MAIN COMMENTS

GC1: The first issue involves the coarse scale GCM predictor variables used by the downscaling model. Using a single GCM (IPSL-CM5A-LR) to calibrate the model and generate simulations is problematic as it leaves the analysis subject to the biases of that individual model (biases identified in "European temperatures in CMIP5: origins of present-day biases and future uncertainties" for example). This GCM has a larger than average climate sensitivity relative to the CMIP5 ensemble and it's response to significantly reduced GHG concentrations may be similarly different from other GCMs. Calibrating a single GCM over a 30-year period should eliminate any biases due to inter-annual or decadal variability but could still influenced by lower frequency modes of variability. Magnitudes of temperature and precipitation in paleoclimate simulations could be amplified or diminished depending on whether the model was fitted in a generally cooler or warmer phase of low-frequency variability. Using an ensemble of models generally limits this effect as well. If only one GCM is feasible, then its characteristics and limitations should be explained in more detail.

GR1: One purpose of the downscaling method, in addition to generating data at a finer grain than generated by the GCM, is actually to correct for the potential biases of the specific GCM. Such correction is possible thanks to the GAM, which acts as a transfer function, and is calibrated using the comparison of the interpolated GCM data with the target CRU data. This point has now been emphasised in the methods (p.4, l.12-20; p.5, l.13-16), where we discuss the biases of the GCM, and in the discussion (p.13-14, l.30-4) It is true that for application of the data to specific issues, an ensemble of models, which would require calibrating a different SDM for each model to correct for the specific bias of each model, would provide more accurate predictions. However, the purpose of this paper is to introduce the method and show how to apply it to a specific model, and such application is therefore out of the present scope. To clarify this point,

and in response to the first comment of reviewer 2, we clarified the objectives of the paper in the introduction (p.3-4, l.22-6)

GC2: The second concern is in the results for Generalized Additive Model (Section 3). There is confusion between the text and Figures 2 and 3 about what is occurring. On Page 8 starting with lines 34-35 and continued onto next page the text states: "Simulated atmospheric temperature at sea level was lower for the LGM than for the present-day period". Is this true? In Figure 2 the legend suggests present-day SLP is lower, while the caption suggests LGM SLP is lower (the figure legend and caption contradict each other over what the solid and dashed lines represent). Further, the domain of the spline for SLP (fitted in present-day) in Figure 3 is 1000 hPa to 1030 hPa which corresponds to the lower valued histogram in Figure 2, contradicting the text. If the spline for SLP in Figure 3 is correct then it implies LGM SLP is described by the solid line and is higher than present-day SLP. This is (hopefully) correct because if the text and Figure 2 caption are correct, the temperature panel would imply that the LGM had high temperatures than present-day, suggesting there is something seriously wrong with the IPSL-CM5A-LR GCM! If the splines of Figure 3 are correct, then a linear extrapolation of the SLP spline into the higher SLP values of the LGM suggests precipitation will have a strong positive response to increasing SLP. This does not seem physically realistic.

GR2: We thank the reviewer for point this error out. We indeed made a mistake when describing the spline of the sea-level pressure, and the interpolation occurs on the right-had extremity of the spline. Please note that the SDM consists of applying a correction to the GCM precipitation using the other atmospheric variables, i.e. to correct the potential biases of the CGM (please see also our response to the previous comment). It does not represent a causal relationship between the predictors and precipitation, and the splines cannot be interpreted separately. In other words, the positive slope of the spline indicates that, according to the comparison with the CRU data, precipitations should be higher at high SLP than they are in the GCM. This point has been

clarified on p.10, l.14-16. Moreover, given the low slope of the spline at this point, and the fact that most pressure values are below ∼1045 hPa, we believe this interpolation will have limited impact on the output of the SDM. The text was modified accordingly on p.10, l.17-19.

GC3: The large differences in temperature in Figure S11 between downscaled and interpolated GCM values also raise doubts about the linear extrapolation of the splines to lower temperatures. The GAM is clearly adjusting the GCM temperatures upward in the majority of the region in response to what appears to be a cold bias in the GCM shown in S12 (the order of subtraction should be specified in both figure captions to confirm this). But does that mean in the LGM the GCM has a 20 degree C cold bias and the GAM is correcting this? Or is the GAM overcompensating and generating temperatures that are too warm because the slope of the temperature spline is too low?

GR3: We agree with the reviewer that the difference in temperature in the North-Eastern end of the study area should be considered carefully (in fact, this point was discussed in the discussion, originally on p.12, l.19-28, now p.15, l.11-11, and see also p.15, l.13-21 on the impact of the calibration area of the results). We believe this difference is a combination of both the underestimation of temperature by the GCM, and an overcorrection of the SDM. However, as we now clarified in the methods (p.6, l.1-8), we are focusing on downscaling the region that was occupied by human populations during the LGM, i.e. mostly Western Europe, South of the ice-sheets. We nonetheless downscaled the whole region to explore in more details the behaviour of the SDM. The paragraph in the discussion was slightly modified to improve clarity, and now reads: "Because the GCM generated reliable temperatures at coarse grain for present-day conditions, which were highly correlated with the CRU present-day temperatures, the three interpolation techniques produced similar linear splines and led to relatively similar values for this variable. The IPSL-CM5A-LR GCM is known to predict lower temperatures than observed at high latitudes in winter (Dufresne et al., 2013).
This bias was indeed observed when comparing the interpolated temperature with the CRU present-day data. As a result, the spline for temperature had a shallow slope at low temperature (Fig. 1). This correction was emphasised for the LGM data generated by the GCM in winter in the North of Europe (Fig. S11), which are outside of the range of present-day temperature, and therefore relied on a linear interpolation of the spline. The large difference in temperature is therefore likely to be a combination of an underestimation of temperature, and an over-correction of the very low temperature by the SDM. However, as stated previously, we are especially interested in downscaling climate data for the region occupied by human populations during the LGM. For the purpose of studying the spatial distribution of modern human population, this overcorrection will have negligible effects, since this region was covered by an ice cap during the time of interest (consequently, no palynological or vertebrate data were available for this region), and the range of values over the whole region in the present-day data encompasses the range of values for the region where humans were present during the LGM (Figs. S2-S7)."

GC4: A useful check of the downscaling model's performance would be to simulate the years in the historical model run (1901-1950 if available, 1951-1960,1990-2005) outside the calibration period and ask how the method performs against CRU observations before attempting to employ the method in time period with substantially different atmospheric forcing conditions. I suggest repeating the figures of S12 and S16 but comparing downscaled values (for winter, summer and using the different interpolation methods) against the CRU observations. If these figures replaced Figures 5 and 8 (moving those to the supplementary figures), it would provide a better picture of the method performance.

GR4: The downscaling performance was validated on the 1950-1960 period. The CRU TS v. 1.2 time series (Mitchell et al. 2004) was used, since it is based on the same methodology used for generating the 1961-1990 climatology used for the calibration and had the same 10 minutes spatial resolution. Since the objective of the work was to

apply the method to LGM data for paleo-anthropological research, we decided to keep figures 5 and 8 (now combined with figures 4 and 7, as figures 5 and 7) in the main text, and to add the validation figures in appendix (Figures S16-S19). The validation was also performed for the kriging technique only, since it is the technique we recommend (now more explicitly in the discussion, p.15, l.8). The results show good agreement for the average temperature and precipitation values, and some small scale variations but overall good agreement in the general spatial patterns for the variability measures.

GC5: It may be beyond the scope of this study but it would be useful to see the GAM fitted separately using proxy data from the 29 sites in past and in present to see how the splines vary between such different climate regimes and whether linear extrapolation is indeed a good assumption.

GR5: We thank the reviewer for this suggestion. However, fitting a GAM over the 29 sites wields several potential issues. Because GAMs are very flexible, using only 29 points may lead to an overfitting of the GAMs, especially when using 6 variables. Moreover, no precise values were available for the past, and we had to rely on reconstructions with confidence intervals, which could be quite large, especially for the BCI technique. We therefore believe that the best way to test the agreement is by comparing the simulated temperature and precipitation with the reconstructions, as we present in Figures 6 and 8.

GC6: The third concern is regarding the comparison of simulated temperature and precipitation against paleo-reconstructions during the LGM. The boxplots of Figures 6 and 9 do not clearly support the claim that the downscaling method is in "good agreement" with the reconstructed values. I suggest removing (or moving to supplementary) the bilinear and bicubic panels and instead display comparisons of annual maximum, minimum and mean values separately for the kriging simulations using proper boxplots. This would provide a clearer comparison between the actual values and allow for at least a visual comparison of the distribution of these values over the 50-year LGM period to be compared. Additionally, how do you measure model performance when the

two selected proxy biomes are significantly different from one another (as occurs more often for precipitation)?

GR6: We have clarified the meaning of the reconstruction ranges from the two methods, which must be interpreted differently (p.8, l.20-24). Given that these reconstructions were generated in independent studies, the corresponding ranges are not directly equivalent to our temperature and precipitation ranges, which are the mean, minimum and maximum values over the 50 downscaled year. The BCI provides a minimum and maximum value over the whole zonobiome, and therefore generates wide ranges. Moreover, that means that simulated temperature and precipitation values close to the extremes of the BCI ranges is expected. The reconstructions by Wu et al. (2007) provide mean temperature of the coldest and warmest months and the ranges are therefore much smaller. However, these comparisons still offer valuable insights to evaluate our data. The meaning of the overlap between the simulated and reconstructed range has also been clarified in the results (p.11, l.10-16 and p.12, l.2-7). Note also that we changed figures 6 and 8 following recommendations from reviewer 2 and now use scatterplots rather than boxplots.

GC7: It would also be particularly useful to evaluate the performance of the GAM in replicating present variability outside the calibration period given the importance of climate variability for human population distributions. Figures S19 illustrates the differences between interpolation methods in the LGM but doesn't show whether the GAM is simulating the variability accurately. It would be useful to see SPI and STI from the GAM compared to the same values for CRU similar to S12 and S16. Further to Figure S19, maps showing the differences between the interpolation methods, as presented in the figures before, would help illustrate the effect of the different methods more clearly. Are the differences in variability from the three methods meaningful and if so are they large enough to suggest the methods could imply different patterns of human migration?

GR7: As explained in our response to comment 4, the downscaling performance was

validated on the 1950-1960 period on the CRU TS v. 1.2 time series. The results show some small scale variations but overall good agreement in the general spatial patterns for the variability measures. Note however that the CRU time series also relies on interpolation techniques (thin-plate smoothing splines) on irregularly space weather stations, and is therefore likely to suffer from its own specific biases. Small scale differences should therefore be interpreted with caution.

All figures presenting the results of the three interpolation techniques were modified to present only kriging and differences between kriging and the other two techniques, as suggested by the specific comment 19 of reviewer 2, to improve clarity and better show the differences between techniques. It is of course difficult to precisely assess the impact of the differences between the three techniques on patterns of human migration, but given that other studies (Burke et al.2014, 2017) found that variability is a key factor governing human distributions, we recommend using the technique providing the best results.

SPECIFIC/TECHNICAL COMMENTS:

SC1: P1 Line 17: Remove the "s" from methods in "Statistical Downscaling Methods".

SR1: This has been corrected.

SC2: P1 Line 27: In the sentence beginning with "Our results" replace "confirming" with "suggesting", add "is" before "suitable" and drop "is sound" at the end. The current sentence is too strong given the evidence presented.

SR2: This sentence has been rewritten.

SC3: P1 Line 31: Replace "their" with "the".

SR3: This has been corrected.

SC4: P8 Line 3: I am skeptical that the p-value for ACO is so low (particularly for temperature) given the sensitivity of the GAM to ACO is so small. If the variance

explained by ACO is indeed statistically significant, the splines and the AIC values would suggest it is not meaningfully significant. This is noted in later paragraphs on this page.

SR4: The fact that the spline is significant is not surprising, given the number of points used for the calibration (the p-values are very sensitive to the size of the dataset, and p-values have been criticised for this, but since they are still the norm, we reported them nonetheless). The flat spline indicates that the effect size of ACO is small, which is a different matter.

SC5: P8 Line 8: The inverse proportionality of temperature to elevation in the GAM spline does not itself imply that the GCM overestimates temperature at high elevation (though it likely does for the reason stated in the next sentence). It merely implies, that in the GAM if elevation increases while the other parameters are constant, then the simulated temperature is expected to decrease.

SR5: This sentence has been rewritten as: "…which means that the coarse-grain temperatures generated by the GCM are higher than observed at fine grain at high elevations".

SC6: P8 Line 16: Sentence beginning with "This should...". The curvature of the lower end of the temperature spline is not negligible so this is not necessarily a safe assumption.

SR6: This sentence was removed, and it now reads: "However, most temperature values in the sites where human presence has been observed during the LGM are within the range of present-day temperature, and the few remaining values are within 10 degrees of the minimum temperature. For very low temperatures during the LGM, the SDM outputs should be interpreted carefully, as we discuss below."

SC7: P10 Line 6: Add "s" to "underestimate".

SR7: This has been corrected.

[Figure]

SC8: P10 Line 30-31: Given the boundary condition issue is present for all the interpolation methods, why not reduce the applicable study area to exclude the outer regions where the downscaled values will be unreliable?

SR8: The size of the area to exclude would vary with the interpolation technique, and is difficult to estimate. For transparency, we therefore decided to provide the full results. Moreover, considering such boundary conditions provides additional details on the differences between the interpolation techniques, which is one purpose of the present work. We added some sentences in the methods (p.6, l.3-8 ) to clarify these points.

SC9: P11 Line 15: "Satisfying results" is subjective, prefer a quantifiable description of how the results compared.

SR9: This sentence was reformulated as: "the method generated results falling within the computed confidence intervals"

SC10: P11 Line 16-16: Sentence beginning with "Elsewhere" seems misplaced here.

SR10: This sentence was reformulated and the new text reads as: "In a separate study, we were then able to test a suite of environmental predictors and demonstrate that climate variability is a key factor governing the spatial distribution of prehistoric human populations during the LGM (Burke et al. 2014, 2017)."

SC11: P11 Line 20: "Critical" is too strong a descriptor here. This study shows the choice of interpolation can reduce spatial artifacts but does not explicitly demonstrate that it alone is most responsible for the GAM accuracy.

SR11: This sentence was reformulated as: "The interpolation technique used in the SDM had a major impact for the spatial patterns of climate variability."

SC12: P11 Line 31-32: "non-linear" One could have linear splines and still end up with differences due to choice of interpolation method.

SR12: This sentence was rewritten for clarification. It now reads: "The splines for

these variables are non-linear and may exacerbate the differences between the bicubic interpolation and the other two techniques."

SC13: P12 Line 12: "Assuming ... accurate". This is not a good assumption and I suggest simply starting the sentence at "We conclude".

SR13: The beginning of the sentence was removed as suggested.

SC14: P12 Line 17: "Reliable temperatures"? There are significant biases in the mountains as shown in Figure S12.

SR14: After rewriting and clarifying the discussion, this sentence does not exist anymore.

SC15: P12 Line 21 starting with "This correction" to the end of the paragraph: Isn't this further evidence that the domain of the study area should be reduced to areas with paleo proxy data and without coverage by an ice sheet?

SR15: The data used to calibrate the SDM must be a compromise between representativity and specificity compared to the area to downscale. In other words, using a region that would only cover the paleo proxys would likely not allow to have representative values for the different climate variables, and using a region that would be too wide would not allow to capture small scale variations. We added details about this point in the methods (p.6, l.1-8) and in the discussion (p.15, l13-21). We also specified that we are nonetheless downscaling the whole region to provide a more complete understanding of how the SDM operates, making clear that using the calibration region presented here is not recommended for downscaling North-East Europe.

SC16: Figure 1 and 3: Add the linearly extrapolated splines in a different colour to show how the variables would respond in regimes that occur during the LGM.

SR16: Unfortunately, this feature is not available in the mgcv package in R, and we did not manage to add the linear extrapolations on the splines. However, we added the range of values for the 12 months over the 50 years during the LGM on the spline and

histogram figures to improve clarity.

SC17: Figure 2: Correct the labelling contradiction between the legend and the caption.

SR17: The legend has been corrected.

SC18: Figure 3: Are the units for the precipitation spline "mm" or "mm/day"?

SR18: The units have been changed to mm/day.

SC19: Figures 6 and 9: Please revise the y-axis ranges of the boxplot figures to span the actual range of data displayed (e.g. there are not any temperature values above 40C yet the plot extends beyond 60C).

SR19: Please note that we changed figures 6 and 8 following recommendations from reviewer 2 and now use scatterplots rather than boxplots.

SC20: Figures 10 and 11: I understand the colour scales here vary from panel to panel to highlight spatial artifacts but it makes interpreting the relative effects of the methods more difficult. I think common colour scales would be more useful given the spatial artifacts should be visible from the contours anyway.

SR20: The colour scale was changed from blue to dark red for temperature. Note that instead of showing all results, we now only show the maps of variability for the kriging technique, and show the difference between kriging and the other 2 interpolation techniques for concision and clarity, as recommended by reviewer 2.

SC21: Figures S1 through S7: There are too many individual panels within these figures and they have insufficient resolution which makes them impossible to read. I suggest presenting only the four seasons for S1, and a few representative panels from the different interpolation methods for S2 -S7 which would allow them to be presented at a readable scale.

SR21: The figures were modified to only show the 4 seasons, and the orientation of the page was changed to landscape to enable better readability of the figures.

SC22: Figures 10 and S17: Please revise the red-black colour schemes to something analogous to the other figures. The large magnitude darker colours obscure the contours and make large areas of the map seem overly homogeneous.

SR22: The colour scale was changed from blue to dark red for temperature. As for the downscaled figures, we now only show the maps of variability for the kriging technique, and show the difference between kriging and the other 2 interpolation techniques for concision and clarity, as recommended by reviewer 2.

SC23: Figures S17 and S18: Add a note in the caption why a different land-sea mask is used in these figures relative to all of the others. I suspect it is because the Mediterranean illustrates the differences in interpolation technique quite well. However, if these are masked out and not used for projections in the LGM it also raises the question of whether these differences are meaningful in the areas actually used in the analysis.

SR23: No land-sea mask had originally been used here because the interpolations are applied before applying the mask in the SDM. However, we agree that this was not coherent, and the mask has been applied to these figures for consistency.

SC24: Figure S19: Specify this is during the LGM. "(STI and SPI values in ]-1,1[)" is a typo?

SR24: Following comments from reviewer 2, the parts of the manuscript referring to the STI and SPI indices have been removed.

ANONYMOUS REFEREE #2

This manuscript deals with the application of a downscaling technique combining interpolation (through three techniques) and General Additive Models (GAMs), over Western Europe during the Last Glacial maximum (LGM). Results are compared to site specific climate proxys from pollen and vertebrate remains data. Its seems well within the scope of Geoscientific Model Development, and deals with the relevant topic of developing statistical downscaling tools that may be used in very different climates like

the LGM. The manuscript needs in my opinion some tightening of the objectives, some work on the clarity of the text and take-home messages, as well as some additional simulation analysis. I detail below these few main comments, together with many specific ones. I can therefore recommend publication of the manuscript only once all these comments are addressed.

We thank the reviewer for these nice comments and for the constructive review he provided.

MAIN COMMENTS

GC1. It is not clear from the start (and down to the choice of figures) what are the objectives of this manuscript. Is it the comparison of downscaling methods (i.e. through different interpolation techniques)? Is it the adequate simulation of reconstructed climate proxy data? Is the target location the whole Europe or only the proxy specific sites? All these questions should be answered from the beginning of the manuscript. As they are currently not answered, the organization of the manuscript and the choice of figures are indecisive (see specific comments below).

GR1. We rewrote the last paragraph of the introduction (p.3-4, l.22-6) to clarify the objectives of the manuscript. As we now state explicitly, we are assessing and refining (comparing the 3 interpolation techniques) the capacity of an SDM method based on a Generalised Additive Model originally designed for the downscaling of climatology data to downscale time series, with a special interest in sites where prehistoric human presence has been recorded. We therefore seek to obtain a good accuracy for the results, while exploring the limitations of the application and acknowledging that results may be improved, for example by using an ensemble of models, as suggested by reviewer 1. We also modified and added some contents in the first paragraphs of the discussion (p.13, l.22-29) to clarify these last points.

GC2. The simulation set-up clearly lacks some present-day validation, as already pointed out by reviewer #1. This would hopefully help disentangling errors/biases from

the interpolation, GAM models, and the driving GCM (see specific comments below).

GR2. The downscaling performance was validated on the 1950-1960 period. The CRU TS v. 1.2 time series was used, since it is based on the same methodology used for generating the 1961-1990 climatology used for the calibration and had the same 10 minutes spatial resolution. Since the objective of the work was to apply the method to LGM data for paleo-anthropological research, we decided to keep figures 5 and 8 (now combined with figures 4 and 7, as figures 5 and 7) in the main text, and to add the validation figures in appendix (Figures S16-S19). The validation was also performed for the kriging technique only, since it is the technique we recommend. The results show good agreement for the average temperature and precipitation values, and some small scale variations but overall good agreement in the general spatial patterns for the variability measures.

GC3. Another consequence of the first main comment above is that a large number of supplementary figures are commented in the main text, which is quite frustrating for the reader. The organization of figures (and associated text) should definitely be redesigned (see specific comments below).

GR3. The figures have been re-designed and re-arranged based on the specific comments below. Given the nature of the work, we had to provide quite a number of figures in supplementary material to allow the reader to investigate some subtleties of the work in more details, while keeping the number of figures acceptable in the main text. Some figures were combined, which should increase the clarity of the manuscript. We also clarified the objectives of the present study, and we think that the current arrangement of the figures is consistent with the logic.

GC4. In relation to the second main comment above, there is little uncertainty discussed in the manuscript, be it a result of the short calibration period for GAMs or from another source like using a single GCM. This should definitely be taken on by the authors for the manuscript.

GR4. The biases of the GCM, which can influence the SDM have now been specified in the Methods (p. 4, l.12-20). In addition, we added a paragraph in the Discussion (p.13-15, l.13-21) in which we discuss in more details some uncertainties related to our results and make recommendation for dealing with them.

SPECIFIC COMMENTS

SR1. P3L1: Please define "taphonomic"

SR1. Taphonomic has been defined in the parenthesis following the word. It now reads "(i.e biases in the fossil record, such as pollen preservation, location of archaeological sites, etc.)".

SC2. P3L23-25: The length of the two GCM simulations is not clear here

SR2. It is now specified that the climate data corresponds to the average of the 1961-1990 period, by contrast with the 50 years time series.

SC3. P4L29: The reference used here for the mgcv R package is not one of those recommended in the citation info of the package. Please correct this.

SR3. The reference has been changed to Wood (2011), as indicated in the citation information of the mgcv package.

SC4. P5L1-2: Is there actually a theoretical reason for the requirement of the same scale? I fully understand the advantage of using e.g. downscaled precipitation as a predictor for local precipitation, but when considering other predictors like SLP, the most informative scale for local precipitation my clearly not be the local scale, but a larger domain shifted in the direction of the prevailing winds (at least in western Europe). This would open quite different approaches for performing this kind of studies that would not require the interpolation step. But this may lead to difficulties given the change in land/sea mask and the presence of ice caps when considering LGM simulations. I would appreciate a comment on that.

SR4. We agree with the reviewer that the extents of the region used for calibration and for the downscaling are important to consider. We now explain in more details in the methods (p.6, l.1-8) that we are especially interested in downscaling the region of Europe where human presence has been observed during the LGM (i.e. South of the ice-sheets), which is why the calibration region encompasses North-East Europe, which has low temperature during the present-day period. However, using a region that is too big would not allow to capture the small-scale variations. We specify that we applied the downscaling on the whole area to better explore how the SDM performs, and added details in the discussion (p.14, l.3-4 and see also p.15, l.16-20) on the risks of using it to downscale North-Europe during the LGM.

SC5. P5L13-15: This starts to be confusing in terms of data. I believe that (for any location) not only the monthly regime (i.e. 12 values only) is used, but the whole 31-year monthly time series. Please be more specific.

SR5. The climatology, i.e. the 30-years average resulting in 12 values for each cell, was indeed used to calibrate the SDM, and then applied to downscale the time series. The climatology was used because the GCM is not precise enough to simulate the temperature and precipitation of a given month in a specific year, which would be require to calibrate the GAM on a time series. However, a GCM can generate temporal patterns for these variables, and we therefore tested the potential of applying an SDM calibrated on a climatology to a time series generated by the same model. These points have been specified in the introduction as we clarified the objectives of the work (p.3, l.21)(cf General Comment 1), and in the methods (p.5, l.29-31).

SC6. FigureS1: This figure is not readable at all. Same for Figures S2 to S7. I would strongly recommend finding a way to make them actually useful.

SR6. The figures were modified to only show the 4 seasons, and the orientation of the page was changed to landscape to enable better readability of the figures.

SC7. P528-P6L3: "Aco" should be defined mathematically in the text without having to

look into Vrac et al. (2007). There is no need to define "Dco" if not used, apart maybe from writing that it is highly correlated to "Aco".

SR7. The mathematical formula of Aco is now provided in Equation 2.

SC8. P6L16: There should be a reference here to Table S1.

SR8. The reference to Table S1 has been added.

SC9. Section 2.5: There should be two additional subsections on the interpolation/downscaling for the present-day reference period (1961-1990) and for a present-day validation period (see Main comment above and comments from reviewer 1).

SR9. A section (2.7) describing the present-day validation outside of the calibration period was added. Please note that we did not include the downscaling of the 1961-1990 climatology, because it would be redundant with the validation on present day data, and would add an unnecessary additional number of figures. Figures S10 and S13 nonetheless compare the interpolated and CRU data for 1961-1990 to discuss the correction performed by the GAM (p.11, l.2-4; p.11, l.27-29), but we considered that this did not deserve a full subsection, that would complexify an already long article.

SC10. P6L24: "Extremes" is a much too strong word here. This set-up (length of time series and temporal resolution) prevents assessing extremes.

SR10. "extremes" has been removed from the text.

SC11. Figure S8: This map should definitely be included in the main text, because this critically shows where to look in European map results. It might also be relevant to systematically indicate these locations in the results maps (depending on their size).

SR11. Figure S8 has been added to the main article and is now Figure 1. The locations were not indicated on the other figures, because they already contain a lot of information, such as the contour line, and adding the site locations would impair their readability.

SC12. P7L18-25: Results on the SPI and STI are not used at all in the manuscript (only in the Supplementary material), so please remove their description (and possible comments) from the main text.

SR12. All parts of the manuscript referring to the SPI and STI have now been removed from the manuscript.

SC13. P7L18-25: The description of SPI computation lacks many important details: (1) what is the climatic norm, i.e. the reference period over which the standardization is based (present-day, LGM) and why? (2) what is the chosen distribution function for monthly precipitation? (3) Is it the same everywhere in Europe? Results are quite sensitive to these issues, as clearly shown in the literature (see e.g. Wu et al., 2005; Stagge et al., 2015)

SR13. All parts of the manuscript referring to the SPI and STI have now been removed from the manuscript.

SC14. P7L18-25: The choice of a variability index as the number of months with SPI between -1 and 1 is actually very strange (and indeed quite irrelevant). The SPI is by definition normally distributed, so the probability of having a SPI between -1 and 1 is 68.27%, which amounts to around 410 months in 50 years (95% confidence interval: 387-432), if the reference period for fitting the precipitation distribution is the same as the computation period (which I believe is the case here, see P7L20). So the spatial pattern observed in Figure S19 is a complete artifact due to (1) the limited length of the period used for fitting the distribution, and (2) the relevance of the specific theoretical distribution used for fitting. Based on the 3 above comments, I strongly suggest removing all the analysis done with SPI/STI.

SR14. All parts of the manuscript referring to the SPI and STI have now been removed from the manuscript.

SC15. Figure 1: It would be great to see the range of present-day and LGM predictors

in these figures in order to directly check statements made in the text P8L12-18.

SR15. The total range of the values is represented by the range of the x-axis of the splines. This is now specified in the figure's caption. In addition, we added grey lines to show the values of temperature at the locations of the archaeological sites in the simulations for all months and 50 years.

SC16. Figure 2: Like reviewer #1, I believe that dotted lines are for the present-day period. Please remove the wrong legend definition from the caption.

SR16. There was indeed an error in the figure caption. The text and figure caption have been modified accordingly, indicating that dotted lines are for present and plain lines for the LGM.

SC17. Section 3.2: As mentioned above for section 2.5, there should be an additional result section for validation the interpolation/downscaling process in a presentday period distinct from the calibration period.

SR17. Section 3.3 has been added to present the results of the validation procedure on a present day time series outside of the period used for calibration.

SC18. Most of results are presented at the annual time scale or for two 3-month seasons. What is then the advantage of fitting GAMs for individual months? I would expect a larger explained variance for annual or seasonal averages. I would appreciate some comments on this issue in the manuscript.

SR18. The maps are presented combining months into seasons to condense the results and increase readability. This has now been specified in the figure captions. Applying the GAM to individual months is nontheless necessary for computing the variability of temperature and precipitation.

SC19. Figure 4: Possible differences between the three interpolation techniques cannot be appreciated from these maps with a common colour scale, because of (1) the large spatial range, and (2) the large seasonal range. Figure 5 looks into all possible differences between the 3 techniques, making both figures relatively redundant. I would therefore recommend choosing one interpolation technique as reference (ideally the one that should be recommended in the conclusion of the manuscript) and plot (1) maps as in Figure 4 for this technique, and (2) differences from this reference with a specific colour scale, as in Figure 5. This would hopefully reduce the number of figures and make the message clearer ("we choose this technique and results with the others are not that different.")

SR19. We thank the reviewer for this advice. Following his recommendation, we combined figures to represent the values for the kriging, and the differences between the kriging and the other 2 interpolation techniques, decreasing the total number of figures in the manuscript and the supplementary material, and making the message clearer.

SC20. Figures S9 to S12. This is a much too high number of figures which shows that work on synthesizing results is clearly lacking. The reader should be presented two things: first, how temperature (and precipitation in a second step) is transformed by the whole downscaling process, through maps of raw, interpolated, interpolated +downscaled, and observations (CRU) in the present-day period. A similar presentation should be made for the validation period, and for the LGM period (for which CRU observations may be replaced by the pollen and vertebrate proxys). This could be made only for the reference interpolation technique. Second, additional maps should show the differences with the two other techniques, possibly through the whole downscaling process. This would require reorganizing figures and text (P9L6-28), but for a much better clarity of the manuscript!

SR20. The number of figures in both the main article and in the supplementary material has been reduced. Since, as we now clarify in the introduction, the purpose of the present work is to explore and refine an existing SDM method design for the downscaling of climatology data to downscale time series of simulated past climate. Since the core SDM method has been described previously (Vrac et al. 2007), we focus on presenting the results and the effect of using the different interpolations in the main

article, since these represent its main contributions. Showing how things change from coarse to interpolated to downscaling is not the main focus here, because the effect of the downscaling will vary depending on the interpolation technique to compensate for the bias the interpolation may induce. Rather, comparisons with the interpolation data is used to shed light on the final results, and such maps are therefore in the appendix.

SC21. P9L25-28, and Figure6: I am not convinced by results as presented here, as these plots are not very appropriate for identifying agreement for each site independently. I would therefore recommend trying scatterplots (with uncertainty bars as here or better uncertainty squares), with reconstructions (BCI, Wu et al. data) on the x-axis and simulations from this paper on the y-axis. The overlap of uncertainty ranges with the diagonal might better inform on the agreement of simulations with reconstructions.

SR21. Following the reviewer's advice, we changed figures 6 and 8 to show scatterplots rather than boxplots. In addition, we clarified the differences between the ranges obtained with the reconstruction methods and the downscaling, and how to interpret the figures (p.11, l.13-16 and p.12, l.2-7).

SC22. P9L30-P10L16: cf. comments on temperature for an additional validation period, revised figure organization, etc.

SR22. Please see our response to previous comments GC2, SC9, SC17 and SC20. We believe that the addition of the present-day validation, the clarification of the objectives and the simplification of the figures make the article clearer and justify its current organisation.

SC23. P10L9-11, "This is due. . . such as precipitation (Wood et al., 2004)": I don't understand why this should lead to the European-scale discrepancies noted in the previous sentence. Please make it clearer.

SR23. This sentence has been re-written as: "This explains the discrepancies between the present-day simulations and the CRU data and, by extension, explains the

adjustments performed by the SDM."

SC24. P10L23-31: I find this paragraph a bit long, compared to other issues elsewhere that would also deserve some explanations.

SR24. This paragraph is a bit long because it requires mathematical explanations, which can hardly be condensed without losing clarity. We clarified the other issues identified by the two reviewers in the rest of the article.

SC25. P11L5-9: As mentioned above, please remove the SPI/STI analysis and results. C6

SR25. All parts of the manuscript referring to the SPI and STI have now been removed from the manuscript.

SC26. P11L25-26: So should we use kriging? Please be more specific on your conclusions about the preferred interpolation method.

SR26. We removed "seems to" on l..., and added the following sentence at the end of the paragraph: "We therefore recommend using kriging for SDM applications based on the method presented here."

SC27. P12L14-15, "more reliable variability": I don't understand. Please make it clearer.

SR27. This part was rewritten as: "generates variability indices with more realistic patterns".

SC28. P12L19-29: Well, this clearly poses the question on whether one should put confidence in GCM outputs at high latitudes (at least. . .). And for this study, this raises the following issue: should the interpolation/downscaling take place over the whole of Europe for reconstructing only a few sites located in the south of the continent. This issue should be seriously taken into account by the authors for the manuscript. Indeed, there may some biases in LGM results in the south due to present-day biases in the

north via the continent-wide GAM modeling. . . I am definitely expecting comments on this potential issue.

SR28. The reviewer is right to point out the issue of the confidence of the results for North-East Europe. We now provide additional details in the methods (p.6, l.1-8) and discussion (p.14, l.3-4 and see also p.15, l.16-20) about this point. As we now clarify, we are especially interested in modelling climate for parts of Europe occupied by human populations during the LGM, therefore excluding North-East Europe, which was covered by an ice cap. However, we applied the SDM to this region in the manuscript to explore in details potential issues with applying this method to a region with data outside of the range of values used for calibration. The manuscript is therefore now more complete, not only showing how to apply the method, but also pointing out potential pitfalls.

SC29. P13L3, "larger-scale patterning": Could you explain and make it clearer?

SR29. This sentence has been removed.

SC30. P13L4-6: I am not sure this sentence is relevant here.

SR30. The sentence has been removed.

SC31. Table 1, "AIC weights": this should be defined and commented in the text.

SR31. The AIC weights have now been defined and commented in the figure caption, to avoid overloading the main text.

Technical corrections 1. P5L25: Remove "interpolated variable"

"interpolated variable" has been removed.

2. P6L29-30: Redundancy of "downscaled, simulated"

"simulated" has been removed.

3. P7L3: Please specify that "bio-climatic indices" is abbreviated as BCI(s) in the

following.

This has been specified.

4. P8L10: font size of "predictor"

The font size has been changed.

5. P11L31: "than for the temperature"

The correction has been made.

---

## Author Response (AR2)

1) In the Introduction, please provide a brief summary, by section (or where necessary sub-section), of the paper. This would be very useful to readers as an overview of the paper structure, so they are prepared when starting to read the full content.

*R: Two paragraphs have been added at the end of the Introduction describing the organisation of the manuscript and the logic behind such organisation. The first paragraph is a summary of the Methods section, and the second paragraph summarises the Results section.*

2) Note that use of a decade-long (1950-1960) period of the CRU TSv.1.2 time series introduces a risk of significant interdecadal internal variability 'aliasing' into the validation procedure. Ultimately, a standard 30-year independent validation period would have reduced this risk somewhat.

*R: The following sentence has been included in the new paragraphs of the introduction (p.4, l.6-8) to specify this point: "Although a longer timeframe could be used to validate the method, potentially reducing the influence of interdecadal internal variability, we validated on an 11-year present-day time series distinct from the calibration period due to computational constraints."*